# Neural representation of abstract task structure during generalization

**Avinash R Vaidya[1]\*, Henry M Jones[1,2], Johanny Castillo[1,3], David Badre[1,4]**

[1]Department of Cognitive, Linguistic, and Psychological Sciences, Brown University, Providence, United States; [2]Department of Psychology, Stanford University, Stanford, Stanford, United States; [3]Department of Psychology and Brain Sciences, University of Massachusetts Amherst, Amherst, United States; [4]Carney Institute for Brain Science, Brown University, Providence, United States

**Abstract** Cognitive models in psychology and neuroscience widely assume that the human brain maintains an abstract representation of tasks. This assumption is fundamental to theories explaining how we learn quickly, think creatively, and act flexibly. However, neural evidence for a verifiably generative abstract task representation has been lacking. Here, we report an experimental paradigm that requires forming such a representation to act adaptively in novel conditions without feedback. Using functional magnetic resonance imaging, we observed that abstract task structure was represented within left mid-lateral prefrontal cortex, bilateral precuneus, and inferior parietal cortex. These results provide support for the neural instantiation of the long-supposed abstract task representation in a setting where we can verify its influence. Such a representation can afford massive expansions of behavioral flexibility without additional experience, a vital characteristic of human cognition.

## Introduction

Many complex tasks we perform daily, though different in their details, share an abstract structure. For example, riding a bike and driving a car differ in many details, including the actions they require, but they share an abstract similarity as modes of transportation. It has long been proposed that humans can learn this abstract structure and leverage it to think creatively, make novel inferences and rapidly generalize knowledge to unique problems we have never encountered before (*Gick and Holyoak, 1980*; *Harlow, 1949*; *Tenenbaum et al., 2011*; *Tolman, 1948*) – e.g. applying rules learned about when to break on a bicycle to an automobile. Reproducing this generativity continues to vex even the most impressive artificial intelligences (*Lake et al., 2017*; *Russin et al., 2020*), and is arguably one of the most defining features of human cognition (*Penn et al., 2008*).

In reinforcement learning, the structure of a task is described by its decomposition into 'states' – variables that specify the condition of the environment (*Sutton and Barto, 2018*). The definition of states is critical to determining how learning takes place, such as whether new information is restricted to a specific set of circumstances or is shared between conditions with overlapping features. States that cannot be resolved from sensory information alone are said to be 'hidden' or 'latent' – in such cases, the organization of these states be must inferred from the distribution of rewards, or other outcomes, throughout the task. In practice, latent states (LSs) have been defined as distributions over the spatio-temporal occurrence of rewards or punishments (*Gershman et al., 2013*; *Nassar et al., 2019*), task stimuli or stimulus features (*Collins and Frank, 2013*; *Gershman and Niv, 2013*; *Tomov et al., 2018*), or conditionalization of action values based on recent task history (*Schuck et al., 2016*; *Zhou et al., 2019*). From this distribution over task features, an agent can infer which conditions might belong to the same LSs and which do not. This information can then be used to segregate or lump together observations, making learning more efficient,

\*For correspondence: avinash_vaidya@brown.edu

and enabling generalization of learning between settings that share the same LSs (*Gershman and Niv, 2010*; *Niv, 2019*).

Recent work has provided evidence that the orbitofrontal cortex (OFC) and hippocampus (HPC) maintain structured, abstract representations of tasks during planning and navigation in conceptual spaces, including latent task states (*Liu et al., 2019*; *McKenzie et al., 2014*; *Schuck et al., 2016*; *Tavares et al., 2015*; *Wilson et al., 2014*; *Zhou et al., 2019*). Likewise, other studies have investigated the neural processes supporting fast acquisition of stimulus-response rules based on latent task knowledge (*Badre et al., 2010*; *Collins et al., 2014*; *Eichenbaum et al., 2020*; *Frank and Badre, 2012*). However, an essential feature of an abstract representation is that it can be used to generalize behaviors to new settings, in absence of feedback, through a process of inference. To date, no study has investigated the neural systems that maintain an abstract task representation that is observed to satisfy this criterion. As a consequence, it remains unknown how the brain carries out this essential function. To address this gap, we used fMRI to test the hypothesis that latent, generalizable task representations used to control our behavior during a task are instantiated in neural activity, particularly in OFC and HPC.

Participants completed a task where they gathered rewards in an environment where a latent, generalizable structure was available. Throughout the experiment, images of trial-unique items from three categories appeared beneath a context, denoted by a scene (*Figure 1*). Participants decided whether to 'sell' each item or pass. If they sold, they would receive or lose reward probabilistically, as determined by the combination of item category and context. Participants saw each of these combinations in short batches of trials termed mini-blocks. Importantly, the contexts could be clustered based on the expected values associated with each item category. This structure provided an opportunity to link these contexts together using an abstract, LS representation based on the distribution of category-value associations over contexts. We hypothesized that participants would form, and later use, this representation to generalize adaptive behaviors to new conditions without any need for reinforcing feedback.

This hypothesis was tested across three phases (*Figure 1—figure supplement 1*). During phase 1, participants learned about the abstract LSs using three item categories (hands, foods, and leaves) across nine contexts (*Figure 1b*). In the next phase, participants learned the values of three new item categories (faces, animals, and objects) in three of the nine contexts; one drawn from each LS (*Figure 1c*). Then, in a final generalization phase, they were tested on the remaining six contexts using the three new image categories, without feedback. Optimal performance depended on generalizing the new category values learned during the second phase to the held-out contexts (*Figure 1d*). We expected that this inference would depend on those contexts being linked to a LS representation within a generalizable task representation based on the expected values of categories encountered during the first phase.

## Results

Participants were tested in three sessions. In session 1, participants completed a behavioral version of the task where performance in the generalization phase determined their inclusion in the fMRI experiment in sessions 2 and 3. 48% of participants passed a criterion of ≥70% accuracy in all 18 generalization conditions and so were recruited for two fMRI sessions. Participants who failed to meet this criterion performed the same task in two additional behavioral sessions rather than in the scanner. Of these participants, 50% ultimately passed the accuracy criterion by the third session. Thus, the majority of participants (75%) could carry out this generalization task given sufficient experience (*Figure 2—figure supplement 1*).

In session 2, all participants carried out the same task, but with new context stimuli – requiring them to learn which contexts linked to which LSs anew. This session also included additional blocks of the generalization phase with conditions pseudo-randomized (rather than organized in mini-blocks). fMRI participants completed these additional blocks in the scanner and the rest of the experiment behaviorally. In session 3, all participants completed a shortened version of the training as a reminder of the task and were presented with new pseudo-randomized generalization blocks (in the scanner for fMRI participants).

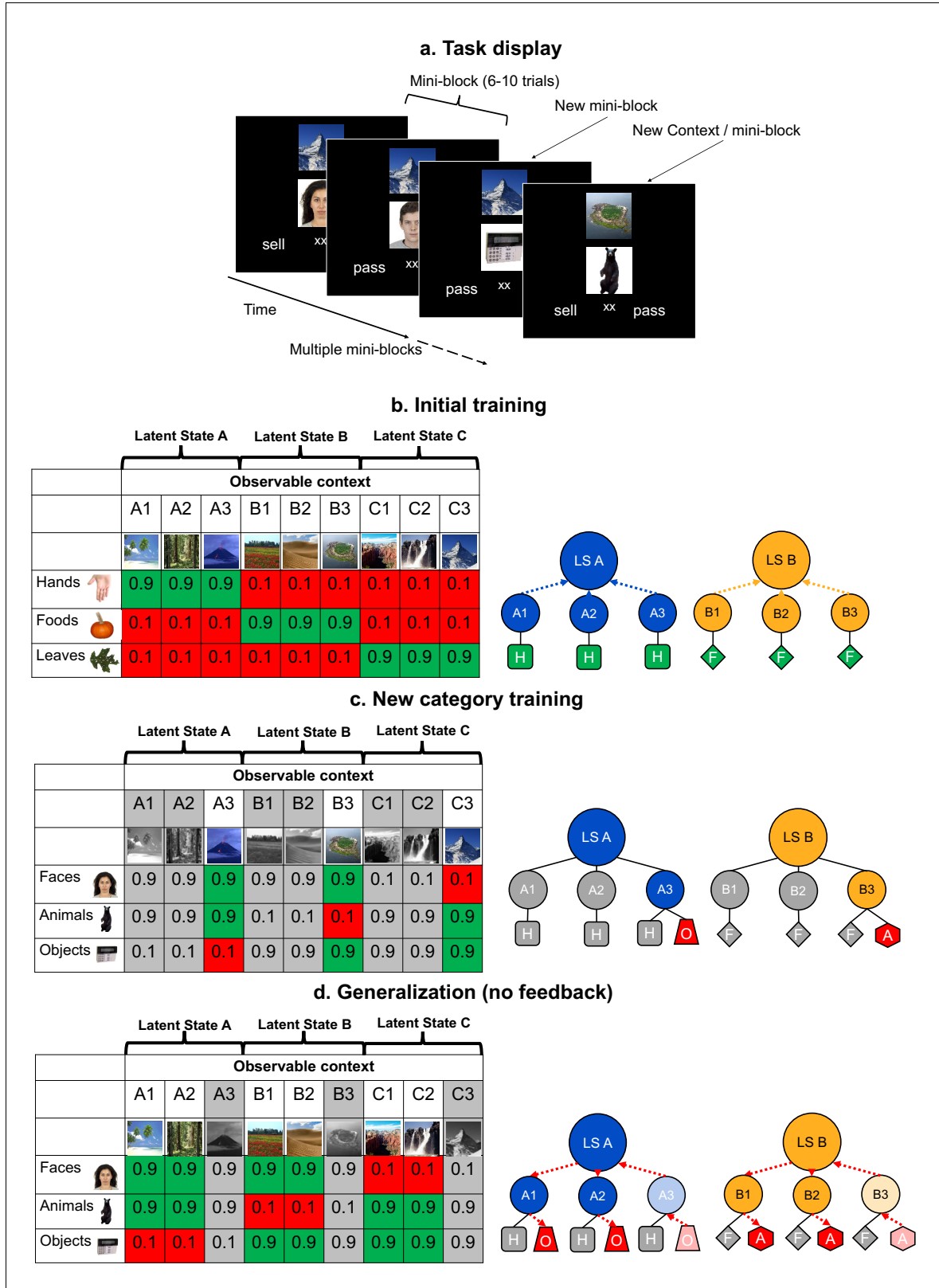

**Figure 1.** Schematic of the experimental task, and its design and logic. (a) In each trial of the training and generalization phases, participants were asked to make a decision to sell or pass on an image, the value of which depended on contexts shown above the image. Throughout the two training phases, participants received feedback on every trial, but not during generalization. Participants saw three categories of images in the same context over small batches of trials for each unique combination (termed a mini-block) before switching to a new context. (b–d). The left tables show the reward

*Figure 1 continued on next page*

*Figure 1 continued*

structure for example context-category pairs across the three phases of the experiment in a single session. Cells show the probabilities of reward for each pair. The right schematics illustrate clustering of contexts by category-values into latent states (LSs; blue and orange arrows) and inference of values via structured knowledge (red arrows). Only two LSs are shown for visualization. Latent State (LS) Hands (H), Foods (F), Objects (O), and Animals (A). (b) In the initial training phase, participants were presented with trial-unique images from three categories of images. These contexts could be grouped together through an abstract LS representation based on the similarity of their category-value associations. (c) Participants were later trained on three new categories in a subset of the previous contexts. Grayed out columns indicate contexts that were left out of this phase. Thus, the values of new categories were trained in only one context in each LS cluster. (d) In the generalization phase, participants were asked to make decisions about the left out, novel context-category combinations without feedback. Participants had to use their knowledge of the LSs linking contexts together learned in the initial training. See *Figure 1—figure supplement 1* for a table laying out the full experimental design over three experimental sessions.

The online version of this article includes the following figure supplement(s) for figure 1:

**Figure supplement 1.** Schematic showing design of whole experiment over multiple sessions.

## Abstract task structure affords inference of novel behaviors without feedback

Analysis of learning curves within mini-blocks of trials demonstrates that participants immediately made use of the abstract task structure in the first segment of the generalization phase. Through the first few trials of each mini-block of the initial and new category training, fMRI participants' accuracy steadily improved. In contrast, in the generalization phase, accuracy was near ceiling from the first trial (mean p(correct)=0.96, SD = 0.06; *Figure 2a*). The rate of this learning curve in the generalization phase was significantly lower than either of the other phases (*Figure 2b*), consistent with participants inferring an adaptive behavior without need for additional experience or feedback (Wilcoxon signed rank tests: $Z$'s $\geq$ 3.21, p's $\leq$ 0.004, Bonferroni-corrected for multiple comparisons). Similar results were observed in participants who completed the behavioral version of the task in session 2 and passed the generalization accuracy criterion (*Figure 2—figure supplement 2a*), with a significantly lower rate of change in the accuracy learning curve in the generalization phase compared to the initial training or new category training phases (Wilcoxon signed rank tests: $W$'s = 45, p's = 0.01, Bonferroni-corrected for multiple-comparisons).

Notably, high accuracy during generalization was accompanied by elevated reaction times (RT; mean = 4.04 s, SD = 1.7 s) on the first trial, with a sharp drop-off subsequently (mean = 1.15 s, SD = 0.08 s; *Figure 2c*). The rate of speeding was significantly steeper in the generalization phase compared to the initial or new category training phases averaged across the first three contexts in each phase (Wilcoxon signed rank tests: $Z \geq$ 3.46, p$\leq$0.002, Bonferroni corrected for multiple comparisons; *Figure 2d*). The same pattern in the rate of speeding was observed in participants from the behavioral group who passed the generalization criterion in session 2 (generalization versus initial training RT rates: W = 36, p=0.047, Bonferroni-corrected for multiple-comparisons; *Figure 2—figure supplement 2d*). Thus, rather than immediately applying a structure learned incidentally during training, participants took additional time to infer values during the first generalization trials – indicative of a concerted, deliberative process.

As each LS was repeated twice via two contexts during generalization, we could test how previously encountering a LS impacted RTs during a repetition. Participants were faster for the second presentation of contexts from the same LS during generalization compared to contexts from different LSs (repeated measures t-tests: (15)$\geq$2.13, p's < 0.05, d's $\geq$ 0.53), indicating that additional inferences were made more quickly when a LS had been accessed previously (*Figure 3a*). Comparison of the rate of exponential curves fit to these RT data in each participant demonstrated that these curves were steeper in the first context compared to the second context averaged across LSs (Wilcoxon signed rank test: Z = 3.36, p=0.0007), but there were no differences in the rate of these curves according to the sequential order of LSs (Wilcoxon signed rank test: $Z \leq$ 1.13, p$\geq$0.2, uncorrected; *Figure 3b*). A numerically similar pattern of results was observed for the rate of change for learning curves in the behavioral group who passed the generalization criterion, but did not reach statistical significance (*Figure 3—figure supplement 1*). These data argue that participants were faster in completing the transfer of new category values to contexts if a context belonging to the same LS representation had already been activated, but did not necessarily use information about other LSs they had encountered in narrowing this inference problem.

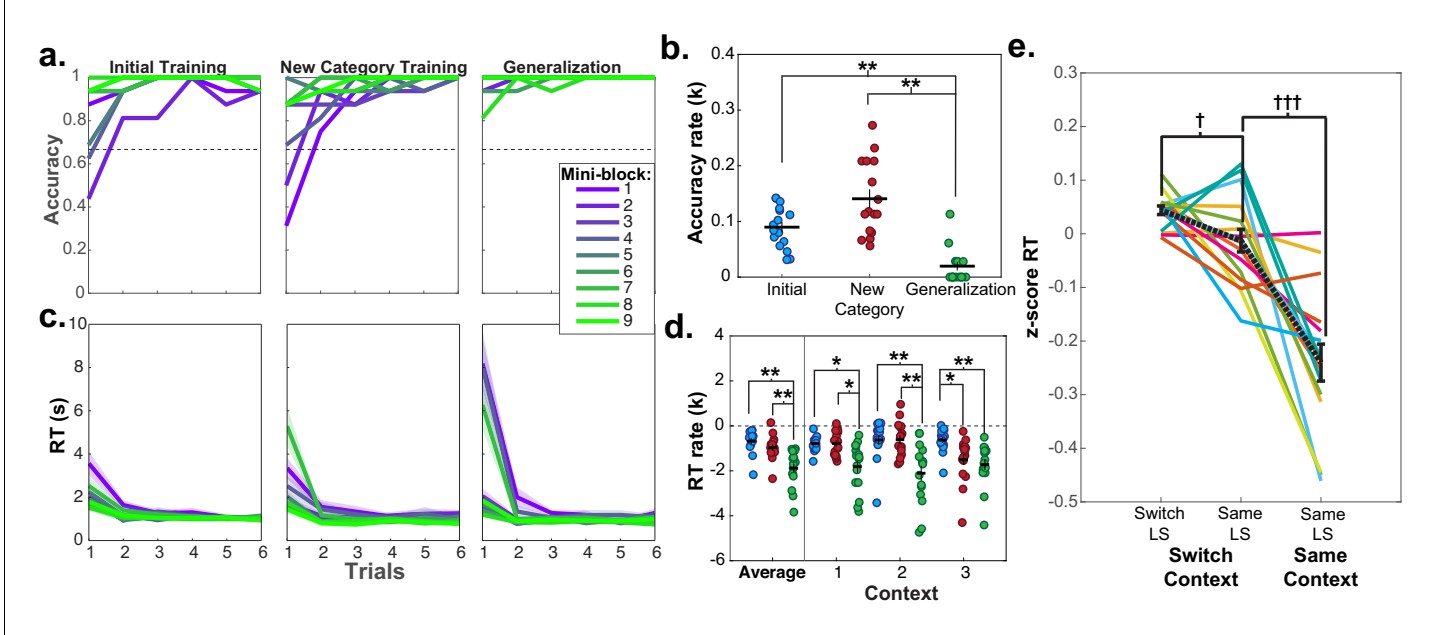

**Figure 2.** Learning curves and repetition effects from training and generalization phases for fMRI participants in session 2. (**a**) Mean accuracy across participants for first six trials of the first nine mini-blocks of initial and new category training, as well as blocked generalization phase. Dashed line indicates chance-level performance. (**b**) Estimated exponential rate of change in mean accuracy in the first six trials for all mini-blocks in each phase. (**c**) Mean reaction times (RT) for each trial in each of the first nine mini-blocks for each of these phases. Shaded area represents standard error of the mean (SEM). Mean and standard error were calculated from the log-transformed RTs. (**d**) Estimated exponential rate of change in RT for first six trials within each phase for the first mini-blocks within the first three contexts in presentation order. For panels (**b** and **d**) each dot represents a single participant, horizontal and vertical bars represent the mean and SEM, respectively. *p<0.05, **p<0.01, Wilcoxon signed rank test, corrected for multiple comparisons. (**e**) RTs for trials from the pseudo-randomized generalization phase from sessions 2 and 3 during fMRI scanning where latent states remained the same or switched from the previous trial, while the context switched or stayed the same. RTs have been log-transformed and z-scored within session for each participant. Each line represents a participant, black dashed line indicates mean, and error bars indicate SEM. †p<0.05, †††p<0.0001, repeated-measures t-test. See *Figure 2—figure supplement 1* for the mean accuracies for the fMRI and behavioral groups for each phase and session of the experiment. See *Figure 2—figure supplement 2* for data from behavioral group participants who passed the generalization accuracy criterion in session 2.

The online version of this article includes the following figure supplement(s) for figure 2:

**Figure supplement 1.** Accuracy in all sessions of the experiment for behavioral and fMRI groups.

**Figure supplement 2.** Learning curves and repetition effects from training and generalization phases for behavioral participants who passed the accuracy criterion for generalization in session 2 (N = 9).

## Latent task states continue to influence behavior after initial inference

After an initial inference, participants may have formed rules for responding to the new context-category combinations and no longer used the LS representation throughout the scanned generalization task. We reasoned that if participants are continuing to utilize the latent abstract task structure, this would be evident in switch costs when the active LS changed between-trials (*Collins and Frank, 2013*). To avoid conflating LS and context switches, we isolated this analysis to only trials where the context changed between trials. We found a significant switch cost in RT for trials when the LS changed versus stayed the same between trials (repeated-measures t-test: *t*(15) = 2.33, p=0.03, *d* = 0.58; *Figure 2e*), which did not differ between scanner sessions (p=0.8). The same effect was independently observed in participants in the behavioral group who passed the generalization criterion (*Figure 2—figure supplement 2*; *t*(8) = 3.45, p=0.009, *d* = 1.15). These data indicate that participants used these LS representations throughout the scanned task. Participants were also faster when both the context and LS remained the same (*t*(15) = 5.29, p<0.0001, *d* = 1.32).

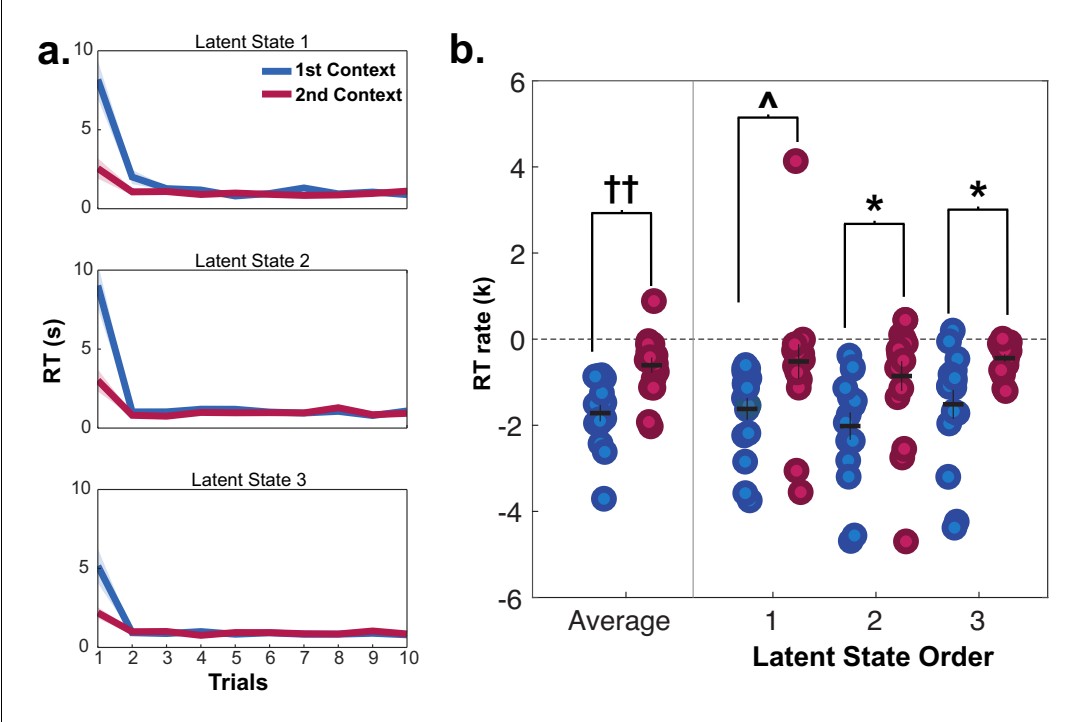

**Figure 3.** Reaction times (RT) for fMRI participants in mini-blocks during generalization in session 2, organized by the presentation of latent states (LS 1, 2, 3) and contexts (first and second). Participants were significantly slower on the first trial of mini-blocks when encountering the first context in a LS compared to the second context in the same LS, but not in subsequent trials. (a) RT curves for mini-blocks organized by the order of LSs and contexts encountered. Lines indicate means, shaded area indicates the SEM. (b) Rates of exponential functions fit to these RT curves. Each dot represents a single participant. $^{\dagger\dagger}p<0.001$, Wilcoxon signed-rank test, $*p<0.05$, Bonferroni-corrected for multiple comparisons, $^{\wedge}p<0.05$, uncorrected. See *Figure 3—figure supplement 1* for data from the behavioral group participants who passed the generalization criterion in session 2.

The online version of this article includes the following figure supplement(s) for figure 3:

**Figure supplement 1.** Reaction times (RT) for participants in the behavioral group who passed the generalization criterion in session 2 (N = 9) during mini-blocks during generalization in session 2, organized by the presentation of latent states (LS 1, 2, 3) and contexts (first and second).

## Generalization behavior is best explained by a hierarchical LS representation

Notably, generalization in this task might not only be achieved through the use of a LS representation to bridge contexts. For example, an associative retrieval process could also explain these results if newly trained category values could be transferred through mediated retrieval of contexts held-out of the generalization phase via shared category-value associations learned in the initial training (*Kumaran and McClelland, 2012*; *Schapiro et al., 2017*).

To test this possibility, we compared four alternative computational models that generated predictions for participants' RTs based on different memory network configurations where contexts, category-values, and LSs were represented as nodes with different levels of activation.

First, the conjunctive associative retrieval (CAR) model represented each condition as a combination of category and context features, and linked these conjunctive representations to each other based on shared features, but had no representation of LS (*Figure 4a*).

The independent associative retrieval (IAR) model similarly did without a LS representation, but rather than only representing contexts and categories as conjunctions, maintained independent representations of each and linked contexts to each other based on shared category-value associations (*Figure 4b*).

In contrast to these associative retrieval models, we included two LS models. A simple LS model connected category-values to three LSs, but forewent any representation of context (*Figure 4c*). A hierarchical latent state (HLS) model connected category-values to contexts, and further clustered

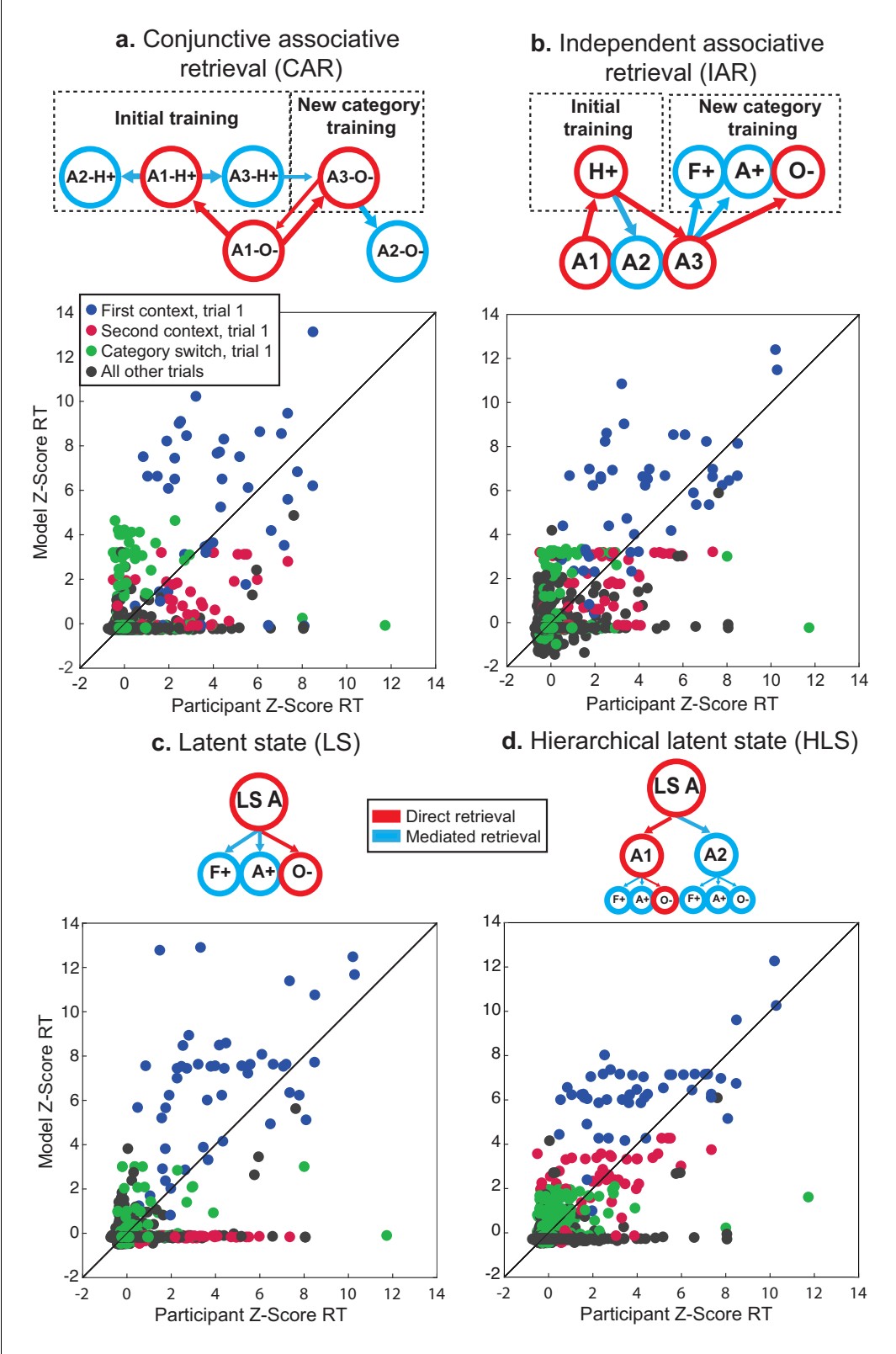

**Figure 4.** Computational modeling of reaction times (RT) during generalization phase for fMRI participants in session 2. Each panel shows simplified schematics for spreading activation in four alternative model networks for a trial involving a decision about an image of an object in context A1 (i.e. A1-O-). (a) Conjunctive associative retrieval (CAR) model where each node represents a context-category conjunction with a particular value. (b) Independent associative retrieval (IAR) model where contexts and category-values are represented separately. (c) Latent state (LS) model where

*Figure 4 continued on next page*

*Figure 4 continued*

category-values are linked to a LS node, without any representation of context. (**d**) Hierarchical LS model with contexts clustered around LSs, and category-values linked to each context. Border and arrow colors signify mode of retrieval. Below each schematic are scatterplots for z-scored reaction times for all participants in all correct trials of the generalization phase and simulated reaction times from each model, color-coded by trial type. See *Figure 4—figure supplement 1* for further details on fit of each model to the data. H, Hands; A, Animals; F, Faces; O, Objects. Values: +, positive expected value; −, negative expected value.

The online version of this article includes the following figure supplement(s) for figure 4:

**Figure supplement 1.** Supplementary analyses for computational modeling of reaction times (RTs).

these contexts around LSs, so that activation of each context and category was inherited from the LS representation (*Figure 4d*).

Each of these models relied on the basic assumption that RTs depend on the degree of activation in the network nodes currently activated. On each trial, activation within the network would increase at different rates depending on the relation of nodes to those that were activated during retrieval. Three rate parameters determined this increase for nodes that were directly activated on a trial ($\alpha_1$), nodes directly connected to those activated ($\alpha_2$), and the rest of the network which might be searched but was not directly relevant to the current trial ($\alpha_3$). To establish that these models differed substantially in their predictions, we tested each model on data simulated using the best fit model parameters from itself and the three alternatives. This cross-model comparison clearly demonstrated that the predictions of these four models were dissimilar and each model provided a much better fit to itself than its alternative counterparts (*Figure 4—figure supplement 1a*).

The HLS model provided the best fit to participants' RT data in the fMRI group in sessions 1 and 2, and the nine participants in the behavioral group who passed the generalization accuracy criterion in session 2 (*Table 1*; *Figure 4—figure supplement 1a*). Comparison of the three rate parameters of this model showed an ordering consistent with the expectation that the degree of activation of items should fall off at a distance from the items presented on the current trial (*Figure 4—figure supplement 1b*), where values for $\alpha_1$ were significantly greater than $\alpha_2$ and $\alpha_3$ (Wilcoxon signed rank test: $Z$'s = 3.51, p's = 0.001, Bonferroni corrected), and values for $\alpha_2$ were significantly greater than $\alpha_3$ ($Z$ = 2.68, p=0.02, Bonferroni corrected).

To understand why the HLS model better accounted for participants' data, we compared RTs simulated from the best fit model parameters for each participant with each of the four models, focusing on data from session 2 for the fMRI group, as it was more pertinent to the neuroimaging results (*Figure 4a–d*). We organized these data by the order of presentation of each context associated with each LS (as in *Figure 3*), switches of category mini-block, and all other trials within a mini-block and then compared the root-mean squared deviation (RMSD) of each model for each trial type. We found that both the LS models provided a better fit on category switch trials compared to the CAR model (repeated-measures t-test, $t(15) \geq 4.76$, p's $\leq 0.002$, $d \geq 1.19$, Bonferroni-corrected for multiple comparisons). Numerically, the HLS model also provided a better fit to these trials than the IAR model, though this difference was not statistically significant ($t(15)$ = 2.02, p=0.06, $d$ = 0.50, uncorrected). The two associative retrieval models tended to systematically overestimate RTs on these category switch trials, likely because category-values associated with the trained context in the new category training phase were only activated by mediated retrieval, whereas in the LS models these

**Table 1.** Sum of negative log-likelihood for four alternative models across participants and fraction of participants where each model was lowest in this measure in parentheses, for each group and session.

| | Conjunctive associative retrieval | Independent associative retrieval | Latent state | Hierarchical latent state |
|---|---|---|---|---|
| fMRI group – session 1 | 3537.78 (1/16) | 3227.38 (2/16) | 3428.23 (0/16) | 3061.04 (13/16) |
| fMRI group – session 2 | 3849.60 (0/16) | 3413.20 (4/16) | 3569.10 (0/16) | 3264.13 (12/16) |
| Behavioral group – session 2 | 2063.65 (0/9) | 1851.76 (0/9) | 1921.61 (0/9) | 1713.73 (9/9) |

The online version of this article includes the following source data for Table 1:

**Source data 1.** Sum of negative log-likelihoods for individual participants in fMRI and behavioral groups for each model in each session.

category-values benefited from inherited direct retrieval of the higher-order LS representation. The HLS model also had lower RMSD for the first presentation of the second context compared to the LS model ($t$(15) = 4.76, p=0.001, $d$ = 1.19, Bonferroni-corrected for multiple comparisons). As the LS model had no separate representation of context, any activation related to a prior LS would be fully transferred to its second presentation through another context, resulting in an underestimation of RTs. Instead, the results were more consistent with separate context representations that inherit activation from a higher-order LSs, that is, the use of a HLS structure.

## Representation of an abstract task representation in multi-voxel activity patterns

Having verified that participants form and use an abstract task representation to control behavior, we sought to test the neural systems that support this representation. We carried out a representational similarity searchlight analysis (RSA) using multiple linear regression to compare empirical representational dissimilarity matrices (RDMs) from pattern activity to hypothesis RDMs quantifying the predicted distances between conditions based on LSs, contexts, item categories, expected value, interactions between these factors, and control regressors (*Figure 5—figure supplement 1*).

We found evidence for a LS representation in bilateral dorsal- and ventrolateral prefrontal cortex (PFC) with a rostral distribution, as well as the bilateral precuneus, left middle temporal gyrus, and bilateral inferior parietal lobules (*Figure 5a*; *Supplementary file 1*). Comparing this statistical map with resting-state functional networks from *Yeo et al., 2011* revealed that this LS representation overlapped most with a frontoparietal network centered on the inferior frontal and intraparietal sulci (*Figure 5—figure supplement 2*).

In contrast, context was associated with activity in the bilateral fusiform gyri (*Figure 5b*). Expected value was associated with activity in OFC, as well as left superior frontal and angular gyri (*Figure 5c*). Item category was robustly represented throughout visual areas and PFC (caudally on the left and broadly on the right; *Figure 5—figure supplement 3*).

To test whether these results were consistent across fMRI sessions, we carried out separate whole-brain searchlight RSAs within each session. The resultant statistical maps for the main effects of interest were similar in both sessions (*Figure 5—figure supplement 4*). Contrasts between statistical maps for all terms in the multiple regression model only revealed a stronger effect for value in the left middle temporal gyrus in session 2 compared to session 3 (MNI coordinates: −48,−22, 0), and no other significant differences in either direction.

As a secondary question, we were also interested in whether higher-order representations of latent task states and expected value influenced each other, or perceptual-semantic representations of item category. To test this, we included regressors for hypothesis RDMs that captured interactions between LSs, value, and item category, as well as control regressors for interactions of these factors with context. Interactions between item category with value and LS were not significant; though items with same value in the same LS were represented more similarly in ventral temporal cortex (VTC) (*Figure 5—figure supplement 1*), observations recapitulated in ROI-based analyses of activity within the VTC (*Figure 6b*).

We also conducted ROI-based RSAs focused on HPC and OFC, given a priori expectation that these regions are involved in representing latent task structure during task performance. RSAs within these ROIs were null for LS representations (*Figure 6a,c*). To test these effects at a finer grain, we conducted second-level tests of the whole-brain searchlight analysis masked by these ROIs. Here, we found signals consistent with LS and expected value representations in central OFC, but not HPC (*Figure 6—figure supplement 1*; *Supplementary file 2*). There were significant effects for category representations in both of these ROIs, in line with broad discrimination between semantically and perceptually distinct categories of stimuli in these regions (*Chikazoe et al., 2014*; *Kuhl et al., 2012*; *McNamee et al., 2013*; *Pegors et al., 2015*).

## Univariate contrast of accuracy

Lastly, we tested a univariate contrast of correct and error responses. Although less functionally specific than the above analyses, we expected this contrast to reveal regions broadly involved in engaging with this generalization task. This contrast revealed greater activation in the HPC and OFC on correct responses (*Figure 7*; *Supplementary file 3*).

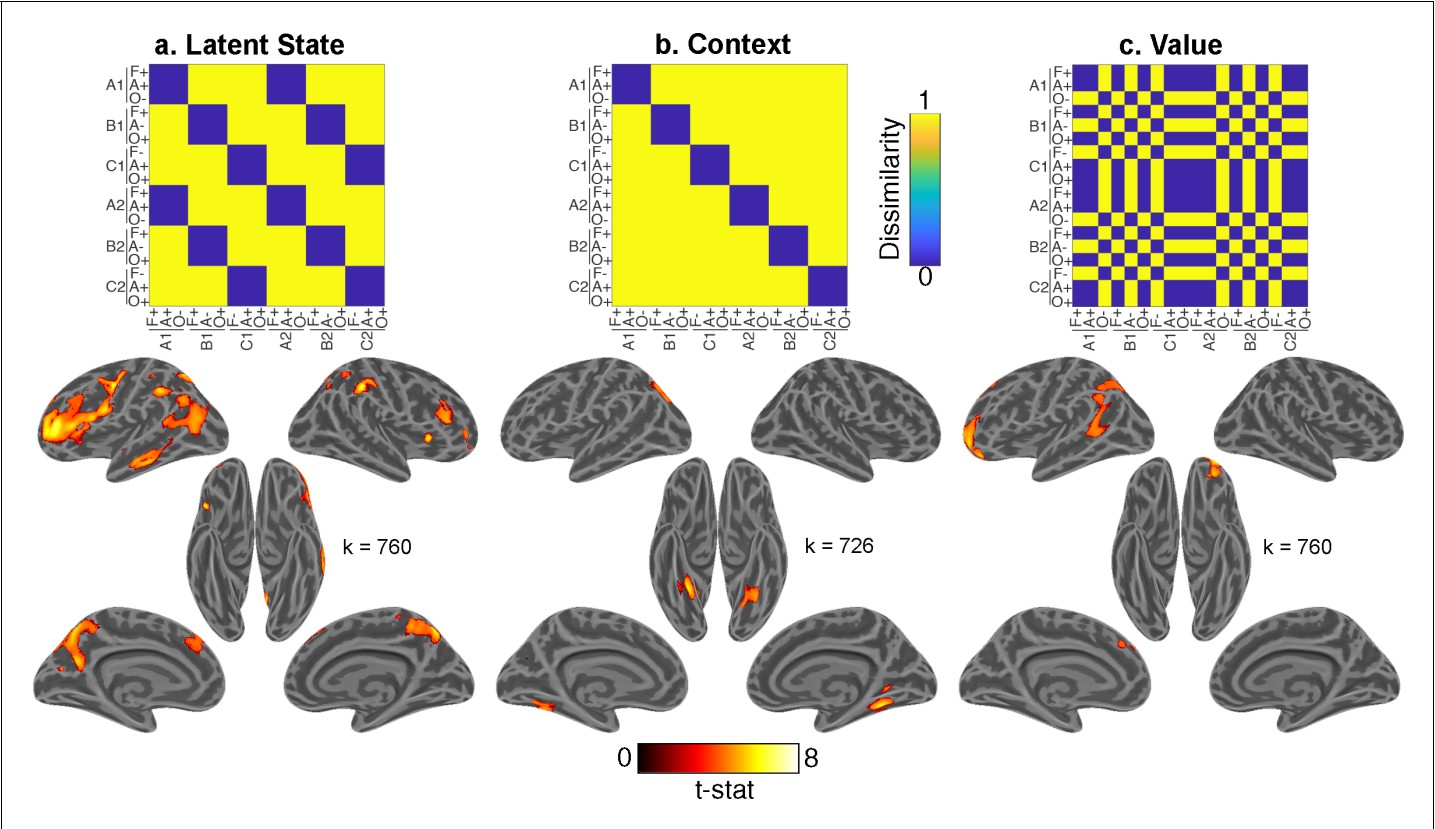

**Figure 5.** Whole-brain representational similarity searchlight analysis for main effects of interest. Each upper panel shows hypothesis representational dissimilarity matrix (RDM) for task factors. Lower panels show t-statistic map from a searchlight analysis testing these predictions in pattern activity projected onto inflated cortical surfaces. All maps are defined with a cluster forming threshold of p<0.001 and corrected for multiple comparisons with permutation tests for defining a cluster extent threshold at p<0.05. These maps include representations of (**a**) latent states (LSs), (**b**) contexts, and (**c**) value. Cluster extent threshold for each contrast is given by the value of k. A, B, C refer to distinct LSs. A1, B1, C1 and A2, B2, C2 refer to distinct contexts that belong to each of those LSs. F, Faces; A, Animals; O, Objects. +, positive value; −, negative value. See *Figure 5—figure supplement 1* for comparison of LS map to 17 network parcellation by *Yeo et al., 2011*, *Figure 5—figure supplement 2* for the statistical map for the item category regressor, and *Figure 5—figure supplement 4* for interaction terms. See *Figure 5—figure supplement 3* for comparison of main effects of interest within each session, and *Figure 5—figure supplement 5* for a view of all regressors included in the multiple linear regression model.

The online version of this article includes the following figure supplement(s) for figure 5:

**Figure supplement 1.** Percentage overlap of latent state (LS) statistical map from *Figure 5a* with 17 network parcellation of functional connectivity networks from *Yeo et al., 2011* based on the data of 1000 participants.

**Figure supplement 2.** Whole brain searchlight analysis for image categories.

**Figure supplement 3.** Statistical maps for main effects of multiple regression model from searchlight representational similarity analyses (RSAs) estimated separately for data from sessions 2 and 3 and projected onto inflated cortical surfaces.

**Figure supplement 4.** Whole-brain searchlight analyses for interaction terms.

**Figure supplement 5.** Hypothesis representational dissimilarity matrices (RDMs).

## Discussion

Here, we observed that a LS representation capable of supporting generalization of knowledge to new settings is instantiated most strongly within activity in mid-lateral PFC and parietal cortex. This pattern overlaps with a network previously implicated in contextual modulation of behavior in hierarchical reinforcement learning and cognitive control tasks (*Badre et al., 2010*; *Choi et al., 2018*; *Collins et al., 2014*). Like hierarchical task structure in those experiments, latent task states enable faster learning, reduced memory load, and greater behavioral flexibility (*Frank and Badre, 2012*; *Koechlin and Summerfield, 2007*). Recent work has similarly implicated mid-lateral and inferior parietal cortex in inferring latent causes within tasks (*Tomov et al., 2018*) and discovering hierarchical rules (*Collins et al., 2014*; *Eichenbaum et al., 2020*). Our findings extend these observations to

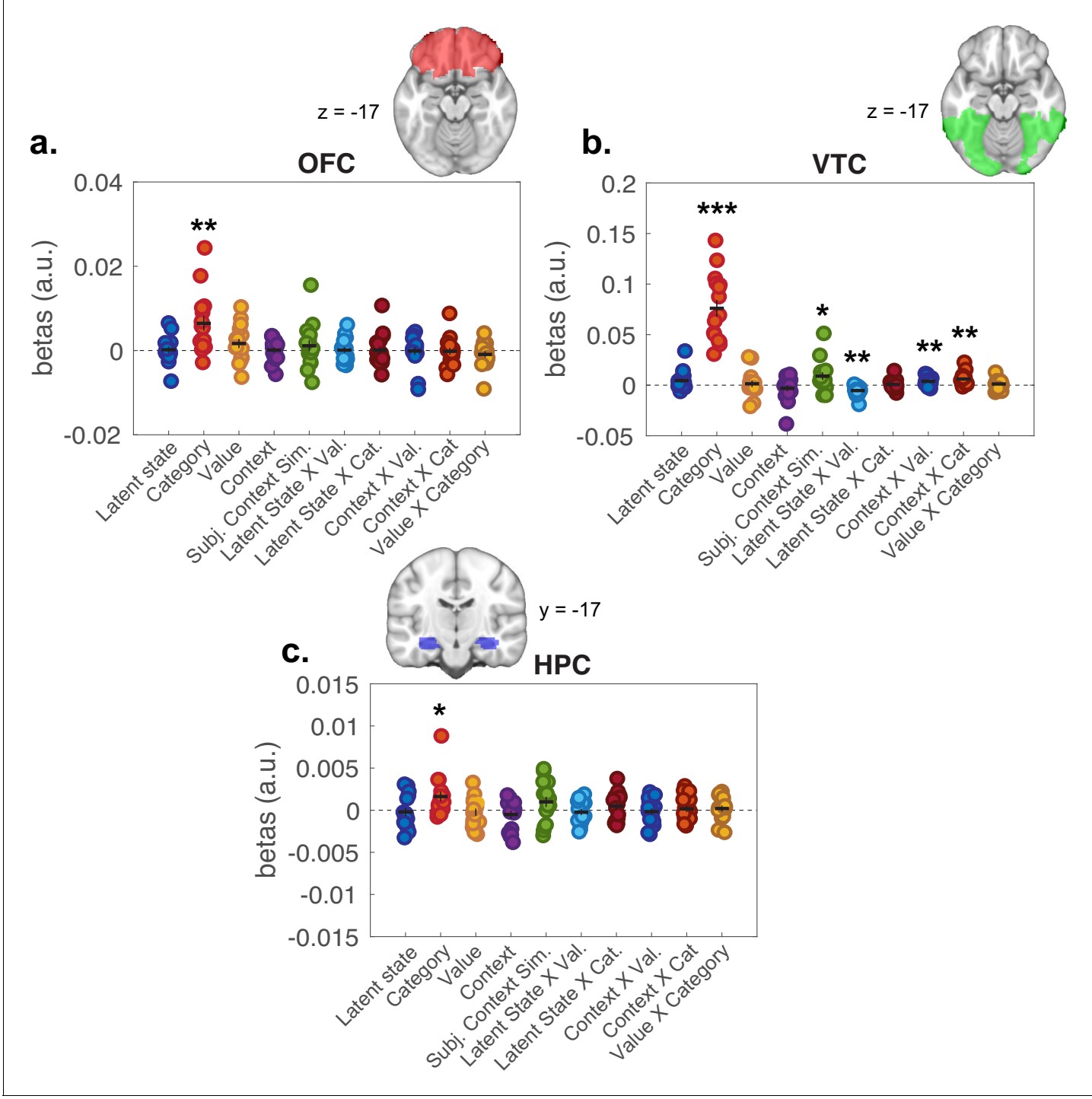

**Figure 6.** Representational similarity analysis results from all voxels included in regions of interest (ROIs). Plots show distribution of beta coefficients across participants from multiple linear regression analyses comparing hypothesis representational dissimilarity matrices (RDMs) with empirical RDMs estimated for each ROI. Each point represents a single participant, with means represented by horizontal bars and SEM as vertical bars. (a) Orbitofrontal cortex (OFC), (b) ventral temporal cortex (VTC), and (c) hippocampus (HPC). *p<0.05, **p<0.01, ***p<0.0001 one-sample t-tests against zero. See *Figure 6—figure supplement 1* for results of second-level tests on value and latent state terms restricted to the OFC ROI.

The online version of this article includes the following figure supplement(s) for figure 6:

**Figure supplement 1.** Results of representational similarity analysis searchlight results with explicit mask in orbitofrontal cortex.

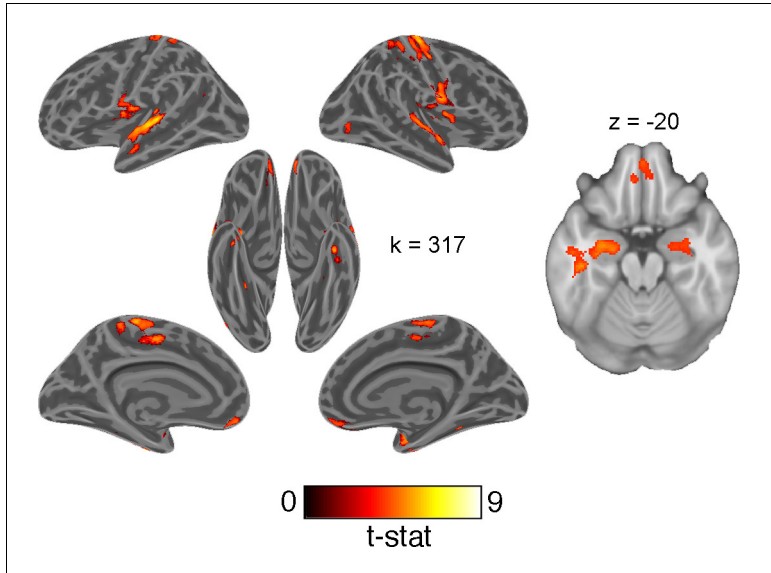

**Figure 7.** Univariate whole-brain contrast of correct and erroneous responses projected onto inflated cortical surfaces and on a single axial slice showing orbitofrontal cortex and hippocampus. This statistical map was defined with a cluster forming threshold of p<0.001 and corrected for multiple comparisons with permutation tests for defining a cluster extent threshold at p<0.05. Cluster extent threshold for each contrast is given by the value of k.

indicate that this network is not just involved in discovering and maintaining task structure, but more generally represents tasks in an abstract format that facilitates novel behaviors, even in the absence of reinforcement. Once formed, such a fully abstract task representation could potentially be used not only for online, stable task control in the face of changing circumstances, but also as the generative mechanism for rapidly generalizing diverse knowledge between settings, affording great behavioral flexibility with minimal training.

Our behavioral results offer insights into the timing and nature of processes that form this abstract task representation. While some accounts of generalization argue that integration of related items occurs during encoding (*Shohamy and Wagner, 2008*; *Zeithamova et al., 2012a*), others have suggested that this can happen online during retrieval through statistical inference (*Kumaran and McClelland, 2012*). In this task, participants could build their task representation at any point prior to the generalization phase, including during the initial training phase. The lengthy RTs during the first trials of generalization suggest that this inference required additional online processing, and this abstract task structure may not have been fully constructed until it was probed. These data are consistent with accounts arguing that generalization during acquired equivalence can also occur at retrieval (*De Araujo Sanchez and Zeithamova, 2020*) and depends on the demand to make new inferences. It is likely that linking of contexts to a LS occurs at both encoding and retrieval, with the timecourse of this process varying across individuals.

Modeling of RTs during the first mini-blocked section of the generalization phase has also provided some insights into the process of how generalization of task information takes place. An associative retrieval model without any LS representation underestimated RTs on category switch trials, as this model did not separate category-values by context or LS. However, a fully integrative representation of LSs was also insufficient, as this alternative collapsed related contexts into the same LS, and underestimated the time needed to make additional inferences about contexts that had not been presented earlier. Instead, our data were most consistent with an intermediate solution where related contexts were bridged by a LS, and category-values were separated by contexts. Thus, our data were consistent with participants accomplishing generalization via an integrated task representation on the basis of shared features (*Schlichting and Preston, 2015*; *Shohamy and Wagner, 2008*; *Zeithamova et al., 2012b*), while still retaining independent representations of each context.

Our task design may have shaped this representation: the instructions strongly encouraged participants to link contexts to an unseen latent variable (see Materials and methods), and the mini-

block trial structure likely encouraged learning the values of each category within each context. Thus, ours is an evidence that we can form such LS representations, use them for generalization, and represent them in frontoparietal network while performing a task based on them. We do not rule out that people cannot also use associative retrieval or that they may do so instead of using a LS in other task contexts. Nonetheless, evidence that we can, and do, form LSs under certain circumstances is an important observation that confirms a long held hypothesis in the field.

Behavioral evidence also indicated that this LS representation was maintained throughout the scanned generalization task. Analysis of RTs showed switch costs for latent task states that exceeded those for changes of context alone. In principle, this representation was not necessary for this portion of the task, as participants could have instead consulted the inferred values of the 18 context-category combinations, or simply memorized which six conditions were associated with negative expected values. Instead, participants appeared to continue referencing this more abstract, and compact, representation of the three latent task states to aid decision-making. Maintaining such a representation would be particularly helpful as a means of compressing task information in memory, and it could be useful in case of further need to update information about these latent task states and propagate this information to linked task conditions (e.g. in a reversal of the expected values for a particular context-category combination). Characterizing if, and how, the neural representation of latent task states described here comes to influence decision-making and learning in such settings remains an important outstanding question.

In this experiment, we were able to cross two factors previously associated with OFC function within the same experiment: latent task states and expected value (*Stalnaker et al., 2015*). We found limited evidence for OFC representing latent task states, though expected value was robustly represented in this ROI. These data are consistent with rodent electrophysiological data indicating that expected value signals contribute more to the variance of activity within this region than representations of task space (*Farovik et al., 2015*; *Zhou et al., 2019*). However, both OFC and HPC were activated for correct responses compared to errors, possibly consistent with their role in retrieving relevant task knowledge, if not in actively representing latent task structure in this experiment.

Prior observations may be reconciled with our results if distinct neural circuits are involved in representing latent task states under different cognitive demands. Much recent work has focused on the involvement of the medial PFC, OFC, and HPC in learning and representing structured knowledge, from the relations of paired associates to complex schemas (*Baldassano et al., 2018*; *Chen et al., 2017*; *Garvert et al., 2017*; *Ghosh et al., 2014*; *van Kesteren et al., 2013*). However, the network identified here as representing latent task states shares greater similarity with regions in lateral PFC and inferior parietal cortex identified by past studies focused on abstract representations of a task in cognitive control settings (*Loose et al., 2017*; *Woolgar et al., 2011*). While these regions belong to mostly distinct functional networks (*Choi et al., 2018*; *Yeo et al., 2011*), information may be shared via strong connections between mid-lateral PFC with medial PFC, OFC, and HPC (*Averbeck and Seo, 2008*; *Petrides, 2005*), and regions that straddle these networks, such as the angular and superior frontal gyri (*Margulies et al., 2009*). Information passed through these select pathways might allow structured knowledge about abstract task elements to inform a control representation in mid-lateral PFC (*Badre and Nee, 2018*).

Frontoparietal systems may be more involved in representing this kind of abstract task structure when it is necessary for actively controlling behavior contingent on changing contexts, as in this experiment. Other work that has examined analogous abstract task representations in the context of conditional action selection and selecting between causal models have found evidence for the involvement of frontoparietal cortex (*Collins et al., 2014*; *Eichenbaum et al., 2020*; *Tomov et al., 2018*), similar to our results. In contrast, OFC and HPC could be engaged more by resolving uncertainty about the prevailing latent task state (*Chan et al., 2016*; *Saez et al., 2015*), or planning within a cognitive map of latent task states (*Schuck et al., 2016*; *Zhou et al., 2019*). Other related work indicates that these regions are involved in, and necessary for, constructing abstract representations (*Kumaran et al., 2009*; *Schuck and Niv, 2019*; *Spalding et al., 2018*; *Zeithamova et al., 2012a*), and so may be particularly engaged by the process of bridging contexts and forming a representation of latent task states. Thus, while OFC and HPC may work together to form an understanding of the latent task structure, mid-lateral PFC and parietal cortex are perhaps more involved in implementing this representation for the purposes of controlling behavior. Indeed, such a division of labor

would be consistent with the long-standing observation of a knowledge-action dissociation from the study of patients with frontal lobe disorders (*Milner, 1964*). As we did not scan participants during training, or the first mini-blocked section of the generalization phase, we cannot yet speak to the neural circuits engaged in initially forming this abstract task representation.

In sum, we have discovered evidence of a neural instantiation of a long-supposed construct of cognitive models: an abstract task representation that enables generalization to new tasks in absence of reinforcement. Future studies could evaluate how and when this representation is formed, when and if this neural representation is necessary for generalization behavior, and why some participants take longer to leverage this abstract task structure. This study thus opens new ground in understanding the neural systems supporting some of the key cognitive processes and behavioral features that distinguish species like humans in their behavioral flexibility and capacity for rapid leaning (*Lake et al., 2017*; *Penn et al., 2008*).

## Materials and methods

### Experimental procedure

The study design and analyses were pre-registered on the Open Sciences Framework prior to collecting the data (https://osf.io/x6fmb). Participants completed an acquired equivalence task where they learned that nine contexts belonged to three different LSs (three per state) on the basis of their shared value associations for three object categories. They then used this knowledge to complete a generalization task where they inferred values for a new set of categories in contexts with which they had not been previously paired. A schematic overview of the structure of this experiment is provided in *Figure 1* and *Figure 1—figure supplement 1*.

To aid in building a structured representation of the task, participants were told they were playing the role of a photographer selling photos of different item categories in three goblin kingdoms with distinct preferences for these categories. In each trial, participants had to make a bet on whether to sell an image or to pass. This decision was a risky choice where selling could be paid off with a reward in fictitious gold or punished with a loss of gold. After a decision to sell, feedback was provided in terms of the gold won or lost. Passing would lead to no change in gold, but feedback was provided in terms of the outcome had the participant chosen to sell. Thus, in all phases of the experiment except for generalization (see below), fully informative feedback was provided independent of participants' choices. Participants were incentivized with a monetary bonus proportional to the amount of gold they earned in the task. All photos were trial-unique so that participants never learned the values of specific images, but of general image categories.

This experiment took place over three sessions. Each experimental session was comprised of three main phases. First, in the beginning of the initial training phase, participants were tasked with selling photos of hands, foods, and leaves in three contexts, represented by a natural scene image above the photo they were selling (A1, B1, C1). Participants were told that the contexts each came from a different 'kingdom', equivalent to the generative LSs (LS A, LS B, LS C), and that each of these kingdoms preferred one category of items strongly over the other two categories. Within each context one item category was associated with a 90% probability of reward and 10% probability of punishment for a decision to sell, while the other two categories were associated with a 10% chance of reward and 90% chance of punishment. In each trial in this initial training phase, participants could gain 50 gold or lose 25 gold for choosing to sell a photo. Thus, the expected value of selling, without knowledge of the probability of reward for each category, was zero.

After two blocks of this training, participants then encountered 'roadwork' after which they would learn about a new context in each kingdom represented by novel natural scenes (A2, B2, C2). This roadwork repeated a second time after another two blocks so that participants learned about three contexts in each of the three kingdoms. Importantly, all contexts associated with a given LS had the same reward probability structure across item categories. Participants were not directly informed of which contexts corresponded to which kingdoms, but had to make these inferences based on their shared category values. The scene images for each context were randomly selected from a set of nine for each participant, so there was no systematic visual relationship between the contexts of each kingdom. Further, as contexts belonging to each 'kingdom' were encountered after each

episode of roadwork, this event bound was not a cue to task structure and no aspect of temporal adjacency could link contexts to a LS.

These initial training blocks consisted of 54 trials with six trials per condition. This initial training phase was organized in batches consisting of a series of trials for each context-category combination (i.e. a 'mini-block'). These mini-blocks were nested by context, so that participants saw each category in the same context one after another (e.g. A1-hands, A1-leaves, A1-foods), before switching to a new context and looping once again through each combination of context-category pairs (e.g. B1-leaves, etc.). The order of presentation for these categories within each context was randomized, as was the order of these contexts. Following these six initial training blocks, participants completed a final reminder training block with the three categories presented in all nine contexts in mini-blocks of eight trials (216 trials total).

In the second phase, participants completed training on new image categories (faces, animals, and man-made objects) within a subset of the previous contexts (e.g. A3, B3, C3). Which set of contexts was presented in this phase (i.e. A1, B1, C1 versus A2, B2, C2 or A3, B3, C3) was randomized across participants. Participants completed 90 trials in the course of a single block, consisting of 10 trial mini-blocks for each combination of the three contexts and three categories.

In the new category training phase, two categories of items were now associated with a 90% chance of reward and 10% chance of punishment in each context, while these values were reversed in one context. The categories that were rewarded and punished in each context were randomized with respect to the categories in the initial training across participants. For example, there was no rule in the experiment that faces have high value in contexts where hands have low value. The punishment associated with selling a photo was increased to 100 gold in this block so that the expected value of selling a photo without knowing the probability of reward in a specific category-context combination was still zero.

Finally, in the third phase, participants carried out a generalization test where they were presented with photos from these new categories in conjunction with the six contexts held out of the prior new category training (e.g. A1, B1, C1 and A2, B2, C2, if A3, B3, C3 were trained with the new categories). Critically, participants did not receive feedback throughout this phase and could thus not learn the values of these new categories from experience. Instead, they would have to rely on their knowledge about which contexts belonged to which kingdoms (i.e. LSs) based on the shared category value associations learned in the initial training phase. This generalization test included 10 trial mini-blocks for each of the 18 conditions (each context-category combination, with three categories and six contexts), resulting in 180 trials total. Participants were informed that the value in gold earned and lost on each trial had been doubled from the new category training phase to further incentivize performance.

It is important to emphasize that generalization during this phase was only possible based on a representation of the shared LSs among contexts. The particular pairings of items and contexts during generalization had never been encountered previously. And, as no feedback was provided during this phase, mappings could not be learned through reinforcement. Further, as the item categories themselves were different from the initial training phase, nothing about the objects or values could link the contexts encountered during generalization to what was learned during phases 1 or 2 except for an abstract LS. That LS already clustered the contexts encountered during generalization with that encountered during new category learning because they had shared a value structure during initial learning, albeit a different value structure on different object categories. Thus, performing accurately during the generalization phase of this task provides unambiguous evidence for reliance on an abstract LS representation.

For all blocks of the training and mini-blocked generalization phases, there was no response deadline. The inter-trial interval (ITI) was generated from a lognormal distribution with a mean of 1 s, maximum time of 4 s, and minimum of 0.5 s. Participants' choice was underlined and the stimuli remained on screen for a 0.5 s inter-stimulus interval. In all blocks, except the generalization test, feedback was displayed for 1.5 s, after which the trial would end. Participants also briefly saw the total gold they had earned within the block after it had ended. However, as there was no way for participants to link these earnings back to the specific trials within the generalization phase, they could not learn the structure of the task from this kind of feedback once they had reached this section of the experiment.

As already noted, this experiment took place over three sessions. The first session was entirely behavioral and served to test how well participants could complete the generalization phase. Prior behavioral pilot experiments had found that approximately 50% of participants did not successfully infer all new context-category values during generalization based on only one experience with training, and so failed to completely learn the latent task structure. As our pre-registered analysis plan depended on participants understanding this structure, we planned, a priori, to exclude from fMRI any participants who did not meet a performance criterion of ≥70% accuracy in all 18 conditions of the generalization test in this first session. Instead these participants were asked to return and complete the rest of the experiment in two additional behavioral sessions. This allowed us to test whether these participants could generalize in this task, given sufficient experience (see main text). That observation helps limit any concerns about generalizability raised by our selection procedures.

In the second and third sessions, after completing the blocked generalization test, participants completed three functional runs of the generalization task but without the nested trial structure (i.e. mini-blocks nested by contexts). These blocks also had a three second response deadline for each trial. In each run, 10 trials from each of the 18 conditions were presented in a pseudo-random sequence optimized for efficiency using Optseq2 with ITIs ranging from 1 to 9 s (180 trials in total) (*Greve, 2002*). This also ensured that the rate of transitions between trials that shared features was effectively at chance (i.e. 33% for LSs, 16.7% for contexts). After each response, the chosen option ('sell' or 'pass') was underlined and the stimulus would remain on-screen until the end of the response deadline. A random subset of six of these 12 optimized sequences was selected for the six functional runs for each participant. Thus, each participant completed 1080 trials of this generalization task over the course of two sessions in the scanner (60 trials per 18 conditions), or behaviorally for those participants who did not pass the generalization criterion in the first session. This dense sampling per participant provided more statistical power for our study by reducing within-subject measurement error. Before completing this task, participants also completed a short practice task with the same mixed trial structure with just one trial from each condition.

Participants were given ample information about the structure of the task in instructions that preceded each new phase of the experiment, but not specific information that about which categories were rewarding in which contexts, and which contexts were part of the same kingdoms (i.e. LSs). Specifically, participants were informed that the outcomes were stable but probabilistic (without being explicitly informed of this probability), and were told how many categories of photos each kingdom preferred in each phase of the task. Participants were also instructed that they would later complete a test phase without feedback, so they should strive to commit the values of photo categories in each context to memory. They were also informed of the latent task structure in a general sense, namely that contexts represented locations within distinct kingdoms, and that the relation of contexts to different kingdoms could be uncovered by their shared values for the item categories, but were never explicitly told which contexts were associated with which kingdoms. Participants were also informed that the contexts within each kingdom could appear to be visually very different from each other to discourage them from using such a strategy to link contexts together. To facilitate forming such an abstract task representation, the instructions specifically suggested that participants use a semantic elaboration strategy that tied together the distinct contexts within each kingdom, and the values of the categories within these contexts. These instructions and the cover story about the goblin kingdoms were included to help encourage participants to learn the abstract structure of the task rather than try and individually commit the values of all 54 context-category conjunctions to memory.

At the end of the third session, outside of the scanner, participants completed a similarity judgment task for the context natural scene images that were shown in their specific generalization test. Participants were shown these images on a black background in random starting positions and asked to use the mouse to drag and drop these images on-screen so that their distances reflected the similarity. Participants were specifically instructed not to use any information about to which kingdoms the contexts belonged, but only the visual content of the image. The Euclidean distances between these images were then used to estimate a subjective, participant-specific estimate of the visual similarity of these images used as a nuisance regressor for RSA analyses.

## Participants

Fifty participants were recruited for this study. Four participants were found to not meet eligibility requirements for the study after session 1 (e.g. they had piercings or implants that were not compatible with MRI), and their data was not analyzed. Twenty-four did not reach criterion performance in the generalization phase (see above) in session 1 and were invited to complete the second and third sessions in behavioral testing settings rather than in the scanner. Six of these participants dropped out of the study before completing both remaining sessions and two did not have complete data due to computer errors. Twenty-two participants passed the generalization criterion and were asked to complete the remaining two sessions in the scanner. Of these participants, two had low accuracy in the generalization phase in the fMRI sessions, indicating a failure to learn the task structure, and their data was not analyzed. Two further participants were excluded because of excessive movement (more than one voxel) in more than one run. One participant had excessive movement only in the last run of the last session and this single run was discarded from analysis. One participant could not complete the experiment due to problems with the scanner on the day of testing, and one participant dropped out of the study before finishing both scanned sessions. In total, 16 participants (10 female, mean age 21.3 years, SD = 3.3 years) passed the generalization criterion on the first day, completed both days of the fMRI experiment and were included in analyses, while 16 participants (11 female, mean age 22.3 years, SD = 2.5 years) did not pass this criterion on the first day and completed the remaining 2 days of testing in behavioral sessions. All participants gave their written informed consent to participate in this study, as approved by the Human Research Protections Office at Brown University, and were compensated for their participation.

The sample size for this study was determined by the target sample size for the fMRI experiment. Data collection was ceased once 16 participants had completed both fMRI scanning sessions while meeting eligibility for inclusion in our analysis (i.e. very little movement and high accuracy during the generalization phase of the experiment). The target sample size was based on a related experiment that successfully used RSA within an OFC ROI (*Chan et al., 2016*). This target sample was halved, as the current experiment involved two sessions of fMRI scanning and we expected within-subject measurement error to be reduced by this dense sampling approach.

## Materials

Eighteen images of natural scenes from *Konkle et al., 2010* were used to represent contexts within kingdoms (nine for session 1 and nine for sessions 2 and 3). These scenes were chosen to be distinct from each other in content and did not include any visible animals, people, or man-made objects (i.e. the image categories included in the fMRI task). Over the course of the experiment, participants saw 360 images in the initial category training for each image category (randomly sampled from a larger set of 468 images). Hand images were taken from the 11 k Hands Database (*Afifi, 2019*), leaf images from the Leafsnap database (*Kumar et al., 2012*), and food images from the Bank of Standardized Stimuli (BOSS) (*Brodeur et al., 2014*), as well as the Foodpics database (*Blechert et al., 2014*). In the new category training and generalization phases, 546 images were used from each stimulus category. These included faces from the Chicago Face Database (*Ma et al., 2015*), animals from the BOSS and CARE databases (*Russo et al., 2018*), and man-made objects, also from the BOSS database. All category images were cropped or padded with white pixels to fit within a square image with a white background.

## Exponential curve fitting

To test the rate of change in RTs and accuracy, we fit an exponential function to these data in the form of

$$y = ab^x$$

where $y$ indicates the predicted trial RT or mean accuracy, $b$ indicates the trial number, and $a$ and $x$ are free parameters that determine the scale and rate of the function respectively. The MATLAB function *fminunc* (Mathworks, Natick, MA) was used to find values for $a$ and $x$ that produced a function that best fit individual participants' data using a least squares cost function. This model-fitting process was run 10 times for each parameter fit for each subject, each time with a different random

starting point drawn from a standard normal distribution. We chose the parameters that resulted in the lowest sum of squares from these 10 iterations to avoid fits that had converged to local minima.

## Computational modeling of reaction times

We developed a set of simple learning models to compare and contrast different explanations for how participants carried out the transfer of option values during the initial generalization phase. The models express retrieval and generalization in terms of spreading activation among a set of nodes that represent the various task elements. In each model, we structured the memory network differently according to the hypothesis tested. However, these models relied on the same basic assumptions and parameters: 1. RTs are a function of the activation strength of the nodes containing retrieved items (*Anderson, 1983*). 2. The activation strength for each item is initialized at zero and increases according to a Rescorla-Wagner learning rule (*Rescorla and Wagner, 1972*), controlled by a single rate parameter bounded by 0 to 1 ($\alpha_1$), each time an option is retrieved, until it reaches an asymptotic value of 1. 3. Activation spreads to contexts and categories linked with retrieved items, controlled by a 'mediated retrieval' parameter ($\alpha_2$). 4. Memory search increases activation in all other task-related items in each trial according to an 'incidental retrieval' parameter ($\alpha_3$). 5. Increases in activation that are inherited from connected nodes falls off according to a power law where the rates of successive steps are multiplied together.

We tested four model variants that each differed in how they represented this memory network, and how activation of task elements propagates during generalization. Importantly, these models are not meant to simulate any particular neural system, but only describe how the format of these representations in memory might differentially affect the retrieval process and, by extension, RTs:

1. CAR – This model implements generalization through associative retrieval by assuming that the memory system stores conjunctions of task-relevant features which can be retrieved if any one of these features is later encountered. This memory network was represented as a matrix of contexts-by-category conjunctions. On each generalization trial, the presented context (e.g. A1) would activate related context-category-value conjunctions from the initial training based on the rate parameter $\alpha_1$ (e.g. A1-leaves- [negative value], A1-food-, A1-hands+ [positive value]). This would lead to the mediated activation of context-category conjunctions with the same value associations from initial training with the rate parameter $\alpha_2$ (e.g. A3-hands+, A2-hands+, etc.), which would in turn lead to activation of conjunctions of new categories associated with the context that appeared in the second training phase (e.g. A3-objects-, A3-Faces +, etc.), as determined by the rate of $\alpha_2^2$. At the same time, the category presented on the current trial (e.g. objects) would activate conjunctions with that category from the new category training phase (e.g. A3-objects-, B3-objects+) at a rate determined by $\alpha_1$ The conjunction shown on the current trial (e.g. A1-objects-) would inherit activation from the related conjunction from the new category training phase (e.g. A3-objects-) and increase in activation at a rate determined by $\alpha_1^2 * \alpha_2^2$. Similarly, the same category conjunction with the held out related context (e.g. A2-objects-) would receive a mediated increase in activation at a rate of $\alpha_1^2 \alpha_2^3$. All other nodes not activated by the current trial would also increase their activation as a rate of $\alpha_3$.

2. IAR – This model also implements generalization through associative retrieval, but rather than relying on episodic conjunctions of context and category, it stores each individual task element independently and links them together based on experience. The memory network was represented as two vectors of contexts and category-value conjunctions (i.e. separate nodes for faces+ faces-, etc.). Each context was linked to their associated category-values from the initial training phase, and only the held-out contexts were directly linked to category-values from the new category training. In each trial (e.g. A1-objects-), activation in the current context (e.g. A1) would increase with the rate parameter $\alpha_1$. This activation would result in direct retrieval of category-values from initial training at a rate of $\alpha_1^2$, and direct retrieval of the context from the new category training phase with the same category-value associations (e.g. A3) at the rate $\alpha_1^3$. Retrieval of this context then led to activation of the relevant category-value node for the current trial (e.g. objects-) from the new training phase at a rate of $\alpha_1^4$, allowing generalization, as well as mediated retrieval of other category-values (e.g. faces+, animals+) at a rate of $\alpha_1^3 * \alpha_2$. Activation of category values from the initial training would also lead to mediated retrieval of the other related context (e.g. A2) at a rate of $\alpha_1^2 * \alpha_2$. All other nodes would increase their activation with a rate set by $\alpha_3$.

3. LS – This model implements generalization through the formation of a LS that replaces separate representations of individual contexts. Here the memory network was represented as a vector of LSs and a matrix of LSs-by-categories, so category-values were separately nested in each LS. In this case, there was no separate representation of a context, so any increase in activation from a trial involving context A1 immediately carried over to context A2. On each trial, activation in the current LS (e.g. LS A) would increase according to the rate parameter $\alpha_1$, and in the current category-value node according to $\alpha_1^2$. All other category nodes linked to the active LS increase in activation according to $\alpha_2$, and all other LSs increase activation according to $\alpha_3$, and their nested categories according to $\alpha_3^2$.

4. HLSs – This model also implements generalization through a LS, but builds on the basic LS model to hierarchically cluster contexts within LSs. Thus, this model builds on the LS model but includes a distinct vector for contexts, as well as a matrix of contexts-by-categories. These contexts and context-category nodes are hierarchically nested within each LS (e.g. contexts A1 and A2 are nested within LS A). Thus, activation for each of these contexts and category nodes is inherited from activation of the LSs and LS-category nodes. On each trial, activation in the current context increases at a rate of $\alpha_1^2$, and in the current context-category at $\alpha_1^3$. Contexts linked to the current LS also increase in activation according to $\alpha_1*\alpha_2$, as these contexts inherit mediated activation from the LS. Categories nested in these contexts increase activation according to the rate $\alpha_1*\alpha_2^2$. Activation in all other LSs increases at a rate of $\alpha_3$, contexts not linked to the current LS increase at a rate of $\alpha_3^2$, and their nested categories according to $\alpha_3^3$.

For each model, predicted RTs were calculated simply as the sum of the activation level in the current context (or LS in case of the third model) and category, subtracted from the maximum asymptotic activation level for both combined. These values were then z-scored and compared to participants' z-scored RTs so that both model predictions and behavioral data were on the same scale. The fit of each of these models was calculated as the negative log-likelihood for a simple linear function. Only correct responses were included in this analysis, as the model assumes correct recovery of option values from memory on each trial. Across all 16 participants in the fMRI group, only 17 trials were errors in session 2, and thus accounted for a very small fraction of the data (0.6%). Parameters were optimized using the MATLAB function *fmincon*, with each model fit for each participant 30 times with random starting points in order to avoid convergence on local minima.

To test whether these models made substantially different RT predictions, we carried out a cross-model comparison using simulated data. We simulated data using the optimized parameters from each participant for each model. We then fit each model on these simulated data, in the same way as with participants' behavioral data and calculated the negative log-likelihood of this fit to test if the generative model provided a better explanation of the simulated data than the three alternatives.

## fMRI acquisition procedures

Whole-brain imaging was acquired using a Siemens 3T Magnetom Prisma system with a 64-channel head coil. In each fMRI session, a high resolution T1 weighted MPRAGE image was acquired for visualization (repetition time (TR), 1900 ms; echo time (TE), 3.02 ms; flip angle, 9°; 160 sagittal slices; 1 × 1 × 1 mm voxels). Functional volumes were acquired using a gradient-echo echo planar sequence (TR, 2000 ms; TE, 25 ms; flip angle 90°; 40 interleaved axial slices tilted approximately 30° from the AC-PC plane; 3 × 3 × 3 mm voxels). Functional data were acquired over three runs. Each run lasted 15.1 min on average (452 acquisitions). After the functional runs, a brief in-plane anatomical T1 image was collected, which was used to define a brain mask that respected the contours of the brain and the space of the functional runs (TR, 350 ms; TE 2.5 ms; flip angle 70°; 40 axial slices; 1.5 × 1.5 × 3 mm voxels). The sequence of scans was identical on both sessions. Soft padding was used to restrict head motion throughout the experiment. Stimuli were presented on a 32-inch monitor at the back of the bore of the magnet, and participants viewed the screen through a mirror attached to the head coil. Participants used a five-button fiber optic response pad to interact with the experiment (Current Designs, Philadelphia, PA).

## fMRI preprocessing and analysis

Functional data were preprocessed using SPM12. Quality assurance for the functional data of each participant was first assessed through visual inspection and TSdiffAna (https://sourceforge.net/projects/spmtools/) and ArtRepair (http://cibsr.stanford.edu/tools/human-brain-project/artrepair-

software.html). Outlier volumes (defined by standard deviations from the global average signal) were interpolated when possible. If interpolation was not possible, a nuisance regressor was added to the run with a stick function at the time points for these volumes. Slice timing correction was carried out by resampling slices to match the first slice. Next, motion during functional runs and days was corrected by registering volumes to the first volume in the first session using rigid-body transformation.

A deformation matrix for spatial normalization to Montreal Neurological Institute (MNI) stereotaxic space using fourth order B-spline interpolation was calculated for the motion corrected functional volumes. The in-plane T1 anatomical image was used to create a brain mask for functional analysis using the Brain Extraction Tool in FSL (https://fsl.fmrib.ox.ac.uk/fsl/fslwiki/). This mask was then normalized to MNI space using the inverse deformation matrix from the normalization of the functional data.

Functional data were analyzed under the assumptions of the GLM using SPM12. Separate regressors were included for correct, erroneous, and missed responses for each condition with the duration of each response set to participants trial-wise RT (or the duration of the stimulus display for missed responses). Nuisance regressors for participant motion (six translational and rotational components) were also included, as was an additional regressor for scan session. Regressors and parametric modulators were convolved with the SPM canonical hemodynamic response function (HRF). Functional data were pre-whitened and high-pass filtered at 0.008 Hz.

## Representational similarity analysis

Whole-brain searchlight and ROI-based RSA were carried out in a two-step process.

First, in participant-level analyses, an empirical RDM was estimated from the cross-validated Mahalanobis distances of the regressors for the 18 conditions (six contexts × three categories) averaged over six runs within the generalization phase from a designated subset of voxels in each participant (either defined by an ROI or a spherical searchlight) using the RSAtoolbox v.2.0 (*Nili et al., 2014*; *Walther et al., 2016*). The lower triangle of this empirical RDM was extracted giving the distances for the 153 condition pairs in this multivoxel space.

Hypothesis RDMs were constructed based on the similarities/dissimilarities that would be expected for a pure representation of a given type. These included five main effects (LS, context, value, image category, and subjective visual similarity of contexts), as well as five interactions (LS × category, LS × value, value × category, context × category, context × value).

In the case of LS, value, and context RDMs, the hypothesized distances were simply set so that conditions that were the same on these factors would have smaller expected distances and larger distances where these conditions were not the same. For the category RDM, hypothesized distances were based on asymmetric distances related to the perceptual-semantic distances of animate and inanimate stimuli observed in many other studies (*Chikazoe et al., 2014*; *Thorat et al., 2019*). Namely, that animate categories (faces and animals) were expected to be more similar to each other than inanimate objects, but inanimate objects were expected to be more similar to animals than to faces. Distances for these hypothesis RDMs were set as ordinal integer numbers to reflect predicted distances (e.g. for the category RDM, the expected distance between two conditions with faces would be 1, the distance between faces and animals would be 2, and between faces and objects would be 4). The RDM for the subjective visual similarity of contexts was specific to each participant and derived from the Euclidean distances of these natural scene images in the similarity judgment task completed at the end of the third session of the experiment. Interaction RDMs were created by extracting main effect RDMs below the diagonal (i.e. the lower triangle of the RDM matrix) as a vector, z-scoring these values and multiplying them together. These interaction regressors allowed us to test where pattern-similarity reflected the modulation of one task component another (e.g. where items belonging to different categories were represented more similarly because of shared value associations).

Second, these hypothesis RDMs were related to the empirical RDM through multiple linear regression analysis, where a coefficient was estimated relating each hypothesis RDM to empirical RDMs allowing us to parcel out variance in representational distances due to multiple factors in the same model (e.g. *Nassar et al., 2019*). Main effects and interaction terms within this model were allowed to compete for variance simultaneously. The lower-triangle of all hypothesis RDMs were extracted as vectors, z-scored, and included as predictors, along with an intercept term, in a multiple

linear regression analysis to calculate beta coefficients relating each hypothesis RDM to the empirical RDM.

## Searchlight analyses

Whole-brain analyses were carried out by passing a spherical searchlight with a radius of 9 mm over each voxel within participants' brain mask in native space. For each participant, beta coefficients for hypothesis RDMs were calculated at each step and averaged over searchlight passes for all voxels included in the searchlight to compute fixed-effects in a first-level analysis. This approach is similar to a common approach of assigning coefficients to a the central voxel of each searchlight and then smoothing these maps before second-level tests (e.g. *Devereux et al., 2013*), but requires one less experimenter degree of freedom in defining the full-width half-maximum (FWHM) of the smoothing kernel. Group level analyses were conducted by normalizing participants' beta coefficient maps to MNI space using the deformation field from the normalization of participants' functional data and computing a one-sample t-test against zero. These volumes were kept in a $1.5 \times 1.5 \times 1.5$ mm space and not resampled. Whole brain t-statistic maps were thresholded at a cluster defining threshold of $p<0.001$ uncorrected. Non-parametric permutation tests (10,000 permutations) were used to derive a cluster extent threshold (k) for each test by randomly flipping the sign for half of participants' contrasts and generating a null distribution based on the suprathreshold maximum cluster statistics. The cluster extent threshold in each contrast was defined as the 95th percentile of this null distribution in order to test for statistical significance at $p<0.05$, corrected for multiple comparisons. (*Nichols and Holmes, 2002*).

## Region of interest analyses

We defined three a priori anatomical ROIs in this study based on prior work using the Automated Anatomical Labelling (AAL2) atlas (*Rolls et al., 2015*). First, the OFC ROI was given the same definition used by a study examining LS representations in this region (*Schuck et al., 2016*), which followed that of *Kahnt et al., 2012* (*Kahnt et al., 2012*). This definition included the bilateral combination of the following regions: the superior orbital gyri, middle orbital gyri, inferior orbital gyri medial orbital gyri, and rectal gyri. The VTC ROI was given the same definition used by a study that showed that pattern activity in this region differentiated between visual images based on animacy (*Chikazoe et al., 2014*), excepting the bilateral parahippocampal gyri. This definition included the bilateral lingual gyri, fusiform gyri, and inferior temporal cortices. The HPC ROI was defined as the bilateral hippocampi. These masks were warped into participants' native brain space using the inverse deformation matrix for all ROI-based analyses. We first conducted an RSA analysis using all voxels within these ROIs by calculating empirical RDMs for the cross-validated Mahalanobis distances from voxels within these ROIs in the same way as in the searchlight analyses, and then using the same multiple linear regression model to relate hypothesis RDMs to these empirical RDMs. These beta-coefficients were then subjected to a second-level, one-sample t-test against zero to estimate statistical significance.

These ROIs were also used as an explicit mask in searchlight analyses to test for effects that were below threshold at a whole-brain cluster-corrected level within HPC and OFC. For these analyses, statistical significance was evaluated as in the whole-brain searchlight analysis with cluster-based permutation tests to compute a cluster extent threshold within these smaller volumes for each contrast, controlling for multiple comparisons at $p<0.05$.

## Univariate analyses

Functional volumes were normalized to a $1.5 \times 1.5 \times 1.5$ mm MNI space and smoothed with an 8 mm FWHM Gaussian isotropic kernel. Beta coefficients for single subject effects were estimated using a fixed-effects model in a first-level analysis. For whole-brain contrasts, these estimates were then included in a second-level analysis using a one-sample t-test against zero at each voxel. As with whole-brain RSA analyses, t-statistical maps were thresholded at a cluster forming threshold of $p<0.001$ and cluster-based permutation tests were used to compute a cluster extent threshold controlling for multiple comparisons at $p<0.05$.

## Comparison with functional networks

The results of the whole-brain LS RSA searchlight was compared with a cortical parcellation based on resting state functional connectivity data from 1000 individuals by *Yeo et al., 2011*. To find the degree of overlap within each of these functional networks, we calculated the proportion of voxels in the cluster-corrected LS statistical map that fell within these 17 networks projected to MNI152 space and defined with liberal boundaries around the cortex.

## Comparison across sessions

To assess the stability of task-relevant representations across scanning sessions, we carried out two separate searchlight RSA analyses using data from within each session, as above. The resultant coefficient maps from the multiple linear regression analysis were then contrasted between sessions in both directions using paired t-tests. These contrasts were then assessed with the same cluster-forming threshold and cluster-based permutation tests to control for multiple comparisons, as in univariate and multivariate analyses.

## Acknowledgements

This work was supported by a Multidisciplinary University Research Initiative award from the Office of Naval Research (N00014-16-1-2832), a Postbaccalaureate Research Education Program grant from the National Institute of General Medical Sciences (R25GM125500), and a Ruth L Kirschstein National Research Service Award to AV from the National Institute of Mental Health (F32MH116592). We thank Apoorva Bhandari and other members of the Badre Lab for constructive discussion and consultation on task design and fMRI analysis, as well as Linda Q Yu for comments on this manuscript. We also thank Emily Chicklis for assistance with data collection.

## Additional information

### Competing interests

David Badre: Reviewing editor, *eLife*. The other authors declare that no competing interests exist.

### Funding

| Funder | Grant reference number | Author |
| --- | --- | --- |
| Office of Naval Research | N00014-16-1-2832 | David Badre |
| National Institute of General Medical Sciences | R25GM125500 | Johanny Castillo<br>David Badre |
| National Institute of Mental Health | F32MH116592 | Avinash R Vaidya |

The funders had no role in study design, data collection and interpretation, or the decision to submit the work for publication.

### Author contributions

Avinash R Vaidya, Conceptualization, Data curation, Software, Formal analysis, Funding acquisition, Validation, Investigation, Visualization, Methodology, Writing - original draft, Project administration, Writing - review and editing; Henry M Jones, Software, Investigation, Methodology, Writing - review and editing; Johanny Castillo, Investigation, Methodology, Writing - review and editing; David Badre, Conceptualization, Supervision, Funding acquisition, Methodology, Project administration, Writing - review and editing

### Author ORCIDs

Avinash R Vaidya https://orcid.org/0000-0001-5423-5655

## Ethics

Human subjects: All participants gave their written informed consent to participate in this study, as approved by the Human Research Protections Office at Brown University, and were compensated for their participation.

## Decision letter and Author response

Decision letter https://doi.org/10.7554/eLife.63226.sa1
Author response https://doi.org/10.7554/eLife.63226.sa2

# Additional files

## Supplementary files

• Supplementary file 1. Activations passing permutation-based cluster correction for whole-brain representational similarity analysis. All reported clusters were significant at the p<0.05, corrected for multiple comparisons after peak thresholding at p<0.001 and permutation-based cluster correction. The critical cluster extent threshold for each contrast is given by the value of k.

• Supplementary file 2. Activations passing permutation-based cluster correction for representational similarity analysis constrained to orbitofrontal cortex region of interest. All reported clusters were significant at the p<0.05, corrected for multiple comparisons after peak thresholding at p<0.001 and permutation-based cluster correction within an explicit mask defining orbitofrontal cortex. The critical cluster extent threshold for each contrast is given by the value of k.

• Supplementary file 3. Activations passing permutation-based cluster correction for univariate contrast of correct and erroneous responses. All reported clusters were significant at the p<0.05, corrected for multiple comparisons after peak thresholding at p<0.001 and permutation-based cluster correction. The critical cluster extent threshold for each contrast is given by the value of k.

• Transparent reporting form

## Data availability

Complete behavioral data from all participants who completed all three sessions of this experiment, un-thresholded statistical maps for whole-brain analyses and beta coefficients for ROI-level analyses have been deposited on the project site for this experiment on the Open Science Framework.

The following dataset was generated:

| Author(s) | Year | Dataset title | Dataset URL | Database and Identifier |
|---|---|---|---|---|
| Vaidya AR, Jones HJ, Castillo J, Badre D | 2020 | Category Betting Task | https://osf.io/g8yj6 | Open Science Framework, g8yj6 |

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
