## [Decision Letter]

**Acceptance summary:**

Combining a clever experimental design with rigorous computational cognitive modeling and neuroimaging, the researchers provide evidence suggesting that latent state structures are encoded by a network of fronto-parietal regions. The bridge between abstract reasoning and adaptive decision-making is an important contribution.

**Decision letter after peer review:**

Thank you for submitting your article "Neural representation of abstract task structure during generalization" for consideration by *eLife*. Your article has been reviewed by Richard Ivry as the Senior Editor, Mimi Liljeholm as Reviewing Editor, and two reviewers. The following individuals involved in review of your submission have agreed to reveal their identity: Charan Ranganath (Reviewer #1); Sebastian Michelmann (Reviewer #2).

The reviewers have discussed the reviews with one another and the Reviewing Editor has drafted this decision to help you prepare a revised submission.

Our expectation is that the authors will eventually carry out additional experiments and report on how they affect the relevant conclusions either in a preprint on bioRxiv or medRxiv, or if appropriate, as a Research Advance in *eLife*, either of which would be linked to the original paper.

Summary:

Vaidya and colleagues present a carefully designed study in order to investigate how the brain represents abstract task structure in a generalization task. Subjects in this study learned "latent states" (i.e., sets of statistically associated scenes) that determined the reward probabilities of objects from different categories. Later, new object categories were introduced and associated with the previously learned task states. Behaviorally, subjects were able to use their knowledge of the abstract task structure to speed new learning and inference on the new object categories. Pattern similarity analyses revealed that abstract task structure, independent from visual information, was represented in left-mid lateral prefrontal cortex, bilateral precuneus, and inferior parietal cortex.

Essential revisions:

All the reviewers thought that this study addresses an important question: Identifying how the brain represents abstract task structure in a generalization task. They agreed that this study uses a very clever behavioral design and a nice combination of behavioral and neuroscientific methods. However, several claims need to be tampered or better justified, and a much more thorough consideration of a broad, heterogeneous and highly relevant literature is warranted.

1) A major concern is that the main claim of the paper – that behavioral and neural data reflect a "latent state" – cannot be evaluated theoretically unless the nature of the latent state representation is carefully and concretely defined. This will require that the authors address the link to prior research on abstract representations of tasks and contexts more broadly, clarifying how they are similar or different in the context of your experiment in both the Introduction and Discussion.

2) As a case in point, the design appears to be one of acquired equivalence (AE) – an extensively studied paradigm in which distinct cues are paired with a common outcome, and subsequent training with one of the cues with a novel outcome generalizes to the second cue during test: that is, given A◊O1, B◊O1 followed by A◊O2, tests reveal expectations of O2 given B. The consensus explanation for such effects is the associative retrieval of O1 by A, and of B by O1, allowing for mediated conditioning of B to O2 during A◊O2 trials. While the current study employs more complex contingencies than those commonly used in studies on AE, it is not clear that some form of associative mediation and retrieval can be ruled out for these effects. The question becomes, then, whether the process or content of complex associative retrieval is what the authors mean by a "latent state representation", or whether they are instead referring to the explicit abstraction of some invariant feature, more in line with previous work on relational reasoning. Either way, the definitions of terms such as "latent state" and "abstract structure" must be clearly laid out and framed in the context of previous relevant work.

3) The authors do not show a clear link between their neural and behavioral results. Some link between these two effects may help strengthen the conclusions that the authors draw. A couple of additional analyses should assess (i) if individual differences in phase 3 accuracy or RT exponential fit relate to subjects latent state representation, (ii) if there is a relationship between switch costs (between latent states) and the fidelity of neural task representation, (iii) To further support that regions in the brain are representing latent states and tracking which state the participant is in, a univariate analysis that identifies regions that are active during latent state switches should be performed. For example, the authors could contrast activity immediately following a latent state switch vs. within a latent state (e.g. trial 1 vs. trial 4 w/in mini block). Is this response bigger when both the context and latent state switches?

4) The latent task RSA analysis revealed a network of regions that is very similar to a network of regions representing "event schemas" across individuals (e.g. Baldassano et al., 2018; Chen et al., 2017). These results should be discussed in a broader context of how the brain represents structure more generally. The supplemental analysis is nice and shows the link to prior work but is not sufficient. The authors do find evidence for "category" representations in a priori ROIs but do not really discuss what the interpretation of these results is. Why is the hippocampus and OFC representing category information during generalization? What does this mean for the field as a whole? For example, it is well established that hippocampus is essential for creating orthogonal representations that are used to disambiguate similar events. How do these results agree or disagree with this interpretation?

---

## [Author Response]

Essential revisions:All the reviewers thought that this study addresses an important question: Identifying how the brain represents abstract task structure in a generalization task. They agreed that this study uses a very clever behavioral design and a nice combination of behavioral and neuroscientific methods. However, several claims need to be tampered or better justified, and a much more thorough consideration of a broad, heterogeneous and highly relevant literature is warranted.1) A major concern is that the main claim of the paper – that behavioral and neural data reflect a "latent state" – cannot be evaluated theoretically unless the nature of the latent state representation is carefully and concretely defined. This will require that the authors address the link to prior research on abstract representations of tasks and contexts more broadly, clarifying how they are similar or different in the context of your experiment in both the Introduction and Discussion.

We agree that this is an important point to address, as a lack of precision can add confusion to the literature regarding this observation. Our revision attempts to clarify how we are using the term “latent state” in the introduction. We define “states” consistent with their use in the reinforcement learning literature: variables that describe the condition of the environment. In the case of a gambling task, for example, the shape or color of a “bandit” could define the state, in other tasks the location on a screen or in a maze might serve the same purpose. States are thus a key substrate for learning, and their formulation dictates how information about the task is shared or separated between different settings. For example, if only the color of the bandit determines the likelihood of reward, it may be useful to represent all bandits of the same color as belonging to a single state, rather than forming separate representations for each bandit by shape and color. This example is a form of generalization, or state abstraction, where information about the value of bandits of the same color is shared across locations, which would speed learning along the color dimension.

Building on this definition, we define “latent states” as states that are not defined by sensory evidence alone, but must be inferred from a distribution of outcomes or other task-relevant information over a set of task features. Thus, the concept “latent state” has been operationalized differently across studies depending on the task at hand. For example, other experiments have defined latent states based on the temporal frequency of tone-shock pairing (Gershman et al., 2013), the spatiotemporal distribution of rewards (Nassar et al., 2019), or action-values conditioned on recent task history (Zhou et al., 2019). In the case of our experiment, a latent state is defined by distribution of rewards and punishments for categories in contexts during the initial training phase. As such, latent states in the present experiment share a similarity to past work examining generalization behavior over latent task sets (Collins and Frank, 2013) or latent causes (Tomov et al., 2018).

The definition of latent state in our task aside, we believe the way latent states are used in our task might provide another account for the difference between our observations and prior reports that have highlighted a role for the hippocampus and OFC in representing latent states. In particular, our task focused on generalization in a phase where there was no external feedback or new information that would allow participants to update their representation of how contexts were related to a latent state. In contrast, other work that focused on active inference of the current latent state from available evidence has found a representation of this information in OFC (Chan et al., 2016). Similarly, other work has focused on latent state representations in OFC and hippocampus that are putatively involved in planning future actions based on recent task history (Schuck et al., 2016; Schuck and Niv, 2019). We discuss this point in the revised manuscript.

2) As a case in point, the design appears to be one of acquired equivalence (AE) – an extensively studied paradigm in which distinct cues are paired with a common outcome, and subsequent training with one of the cues with a novel outcome generalizes to the second cue during test: that is, given A◊O1, B◊O1 followed by A◊O2, tests reveal expectations of O2 given B. The consensus explanation for such effects is the associative retrieval of O1 by A, and of B by O1, allowing for mediated conditioning of B to O2 during A◊O2 trials. While the current study employs more complex contingencies than those commonly used in studies on AE, it is not clear that some form of associative mediation and retrieval can be ruled out for these effects. The question becomes, then, whether the process or content of complex associative retrieval is what the authors mean by a "latent state representation", or whether they are instead referring to the explicit abstraction of some invariant feature, more in line with previous work on relational reasoning. Either way, the definitions of terms such as "latent state" and "abstract structure" must be clearly laid out and framed in the context of previous relevant work.

This is an insightful point that raises the question: Do participants integrate contexts into a representation of a latent task state to help complete the generalization task, or do they generalize through a process of associative retrieval? We believe this is important to resolve because a mediated associative retrieval account would suggest that an integrated task representation is not used to carry out generalization, and it would have different implications for the interpretation of our functional imaging results. To distinguish between these alternatives, we conducted a computational modeling analysis to formalize these hypotheses and compare how well they account for participants’ reaction time data in the first mini-blocked section of the generalization phase.

We developed a set of simple learning models to compare and contrast different explanations for how participants carried out the transfer of option values during the initial generalization phase. The models express retrieval and generalization in terms of spreading activation among a set of nodes that represent the various task elements. In each model, we structured the memory network differently according to the hypothesis tested. However, these models relied on the same basic assumptions and parameters: (1) Reaction times are a function of the activation strength of the nodes containing retrieved items (Anderson, 1983). (2) The activation strength for each item is initialized at zero and increases according to a Rescorla-Wagner learning rule (Rescorla and Wagner, 1972), controlled by a single rate parameter bounded by 0 to 1 (α_1_), each time an option is retrieved, until it reaches an asymptotic value of one. (3) Activation spreads to contexts and categories linked with retrieved items, controlled by a ‘mediated retrieval’ parameter (α_2_). (4) Memory search increases activation in all other task-related items in each trial according to an ‘incidental retrieval’ parameter (α_3_). (5) Increases in activation that are inherited from connected nodes falls off according to a power law where the rates of successive steps are multiplied together.

We tested four model variants that each differed in how they represented this memory network, and how activation of task elements propagates during generalization (see Figure 4). Importantly, these models are not meant simulate any particular neural system, but only describe how the format of these representations in memory might differentially affect the retrieval process and, by extension, reaction times:

1) Conjunctive associative retrieval – This model implements generalization through associative retrieval by assuming that the memory system stores conjunctions of task-relevant features which can be retrieved if any one of these features is later encountered. This memory network was represented as a matrix of contexts-by-category conjunctions. On each generalization trial, the presented context (e.g. A1) would activate related context-category-value conjunctions from the initial training based on the rate parameter α_1_ (e.g. A1-leaves- (negative value), A1-food-, A1-hands+ (positive value)). This would lead to the mediated activation of context-category conjunctions with the same value associations from initial training with the rate parameter α_2_ (e.g. A3-hands+, A2-hands+, etc.), which would in turn lead to activation of conjunctions of new categories associated with the context that appeared in the second training phase (e.g. A3-objects-, A3-Faces+ etc.), as determined by the rate of α_2_^2^. At the same time, the category presented on the current trial (e.g. objects) would activate conjunctions with that category from the new category training phase (e.g. A3-objects-, B3-objects+) at a rate determined by α_1_. The conjunction shown on the current trial (e.g. A1-objects-) would inherit activation from the related conjunction from the new category training phase (e.g. A3-objects-) and increase in activation at a rate determined by α_1_^2^*α_2_^2^. Similarly, the same category conjunction with the held out related context (e.g. A2-objects-) would receive a mediated increase in activation at a rate of α_1_^2*^α_2_^3^. All other nodes not activated by the current trial would also increase their activation as a rate of α_3_.

2) Independent associative retrieval – This model also implements generalization through associative retrieval, but rather than relying on episodic conjunctions of context and category, it stores each individual task element independently and links them together based on experience. The memory network was represented as two vectors of contexts and category-value conjunctions (i.e. separate nodes for faces+ faces-, etc.). Each context was linked to their associated category-values from the initial training phase, and only the held-out contexts were directly linked to category-values from the new category training. In each trial (e.g. A1-objects-), activation in the current context (e.g. A1) would increase with the rate parameter α_1_. This activation would result in direct retrieval of category-values from initial training at a rate of α_1_^2^, and direct retrieval of the context from the new category training phase with the same category-value associations (e.g. A3) at the rate α_1_^3^. Retrieval of this context then led to activation of the relevant category-value node for the current trial (e.g. objects-) from the new training phase at a rate of α_1_^4^, allowing generalization, as well as mediated retrieval of other category-values (e.g. faces+, animals+) at a rate of α_1_^3^*α_2_. Activation of category values from the initial training would also lead to mediated retrieval of the other related context (e.g. A2) at a rate of α_1_^2^*α_2_. All other nodes would increase their activation with a rate set by α_3_.

3) Latent state – This model implements generalization through the formation of a latent state that replaces separate representations of individual contexts. Here the memory network was represented as a vector of latent states and a matrix of latent states-by-categories, so category-values were separately nested in each latent state. In this case, there was no separate representation of a context, so any increase in activation from a trial involving context A1 immediately carried over to context A2. On each trial, activation in the current latent state (e.g. latent state A) would increase according to the rate parameter α_1_, and in the current category-value node according to α_1_^2^. All other categories nodes linked to the active latent state increase in activation according to α_2_, and all other latent states increase activation according to α_3_, and their nested categories according to α_3_^2^.

4) Hierarchical latent states – This model also implements generalization through a latent state, but builds on the basic latent state model to hierarchically cluster contexts within latent states. This model was equivalent to the structure assumed *a priori* in our original submission (see schematic in Figure 1). Thus, this model builds on the latent state model but includes a distinct vector for contexts, as well as a matrix of contexts-by-categories. These contexts and context-category nodes are hierarchically nested within each latent state (e.g. contexts A1 and A2 are nested within latent state A). Thus, activation for each of these contexts and category nodes is inherited from activation of the latent states and latent state-category nodes. On each trial, activation in the current context increases at a rate of α_1_^2^, and in the current context-category at α_1_^3^. Contexts linked to the current latent state also increase in activation according to α_1_*α_2_, as these contexts inherit mediated activation from the latent state. Categories nested in these contexts increase activation according to the rate α_1_*α_2_^2^. Activation in all other latent states increases at a rate of α_3_, contexts not linked to the current latent state increase at a rate of α_3_^2^, and their nested categories according to α_3_^3^.

For each model, predicted reaction times were calculated simply as the sum of the activation level in the current context (or latent state in case of the third model) and category, subtracted from the maximum asymptotic activation level for both combined. These values were then z-scored and compared to participants’ z-scored reaction times so that both model predictions and behavioral data were on the same scale. The fit of each of these models was calculated as the negative log-likelihood for a simple linear function. Only correct responses were included in this analysis, as the model assumes correct recovery of option values from memory on each trial. Across all 16 participants in the fMRI group, only 17 trials were errors in session 2, and thus accounted for a very small fraction of the data (0.6%). Parameters were optimized using the MATLAB function *fmincon*, with each model fit for each participant 30 times with random starting points in order to avoid convergence on local minima.

To test whether these models made substantially different reaction time predictions, we carried out a cross-model comparison using simulated data. We simulated data using the optimized parameters from each participant for each model. We then fit each model on these simulated data, in the same way as with participants’ behavioral data, to test if the generative model better fit the simulated data than the two alternatives (Figure 4 —figure supplement 1D). This analysis confirmed that these models made distinct predictions with little confusion between models.

To briefly summarize the results, the hierarchical latent state model provided the best fit to the reaction time data for a large majority of fMRI participants in sessions 1 (13/16) and 2 (12/16), as well as all of the behavioral participants who passed the generalization criterion in session 2 (N = 9; Table 1). We did not examine session 1 of the behavioral group, as they failed the generalization criterion in this session — and our models assumed retrieval of the correct response. We also did not examine data from session 3, as participants had already completed the same generalization task in session 2 and this session served only as a reminder.

Examining the data of fMRI participants in session 2 more closely, we found that, compared to the hierarchical latent state model, the associative retrieval models overestimated RT on category switch trials, likely because category-values associated with the current context only increased activation through mediated retrieval in these two models. In contrast, in the latent state models, these nodes inherited activation from direct retrieval of the latent state. However, the (non-hierarchical) latent state model primarily underestimated reaction times on the presentation of a second context, as this model did not have an independent representation of context from latent state and carried over this activation from the previously presented context. Thus, the hierarchical latent state model that maintained separate representations of contexts clustered around each latent state provided the best fit to these data as it was able to capture both category shifts and responses to the second context in line with participant behavior (see Figure 4—figure supplement 1).

In the revised manuscript, we describe the results of this modeling analysis in the Results section and a description of the analysis and model fitting in the Materials and methods section. We believe that this analysis makes an important new contribution by clarifying the nature of latent state generalization and explicitly formalizing and testing how different formats for this representation would influence behavior.

We believe that these results support our claim that participants used a latent state representation to complete this generalization task. However, we cannot speak to whether participants form such a representation obligatorily to complete acquired equivalence tasks in general, or if this finding reflects specific characteristics of our task. In particular, task instructions emphasizing the existence of “goblin kingdoms” (i.e. latent states) to which contexts were linked, and the mini-blocked structure of the generalization task and training (where categories were presented sequentially in each context), may have encouraged the use of such a hierarchically organized latent state representation. The kind of representation used to complete acquired equivalence tasks remains a point of some debate, and may depend on task demands (De Araujo Sanchez and Zeithamova, 2020; Shohamy and Wagner, 2008). We now raise these points in the Discussion.

It is also worth noting that this modeling analysis focuses on the mini-blocked generalization phase, not the scanned mixed generalization phase that was the focus of our fMRI analyses. In principle, it would be possible for participants to use a different representation to complete this task after having completed the initial generalized (e.g. memorizing the values of the 18 new context-category combinations, or just the six items with negative expected values). However, we found participants had RT switch costs for latent states that exceeded those switch costs for context alone in this phase of the experiment. These data thus argue that participants continued to refer to this latent state representation throughout the scanned generalization phase. We discuss this point in the Discussion section.

3) The authors do not show a clear link between their neural and behavioral results. Some link between these two effects may help strengthen the conclusions that the authors draw. A couple of additional analyses should assess (i) if individual differences in phase 3 accuracy or RT exponential fit relate to subjects latent state representation, (ii) if there is a relationship between switch costs (between latent states) and the fidelity of neural task representation, (iii) To further support that regions in the brain are representing latent states and tracking which state the participant is in, a univariate analysis that identifies regions that are active during latent state switches should be performed. For example, the authors could contrast activity immediately following a latent state switch vs. within a latent state (e.g. trial 1 vs. trial 4 w/in mini block). Is this response bigger when both the context and latent state switches?

We agree that a clear link between the neural evidence for a latent task state representation and behavior would be helpful in clarifying the role of this representation in this task. However, our experiment was not designed for the purpose of detecting such a relationship, and is ill-suited for testing these possibilities, as we explain below. We have pursued each line of analysis suggested by the reviewers, but did not find any result that we felt confident including in the revised manuscript. Nonetheless, we believe that the main results of our manuscript demonstrating evidence of a latent state representation that is generative and demonstrably abstract during generalization are important and worth reporting. We believe that the question of how this representation impacts behavior will ultimately require more work and future experiments designed to answer this question.

Here we address each of these suggested analyses in turn below:

i) If individual differences in phase 3 accuracy or RT exponential fit relate to subjects latent state representation

We took a dense-sampling approach in this experiment, collecting two sessions of fMRI data for a relatively small number of participants (N=16). We expect that we would thus be underpowered to detect any effect driven by between-subjects variance, and would be skeptical of any correlation across these 16 participants in neural and behavioral data.

However, for exploratory purposes, we extracted β coefficients for the latent task state representation from an ROI defined by voxels that exceeded the permutation-based cluster-correction threshold for statistical significance at P < 0.05 (Figure 5). There was a moderate, but not statistically significant, negative correlation between this neural measure and the rate of the exponential function fit to participants’ accuracy (ρ = -0.347, *P* = 0.18, Spearman correlation), and no relationship between the mean exponential rate for reaction times for the first context in each latent state (r = -0.08, *P =* 0.77, Pearson correlation). Please see Author response image 1 (left, reaction time (RT) rate; right, accuracy (ACC) rate):

Visual inspection of the data is suggestive of a trend towards a negative correlation for both the RT and accuracy rates, perhaps offset by one or two participants. However, given the small sample size, we are not confident enough in these results to present them in the manuscript.

ii) If there is a relationship between switch costs (between latent states) and the fidelity of neural task representation

We agree that this is an interesting and important question. However, we used a rapid event-related design that was optimized for the estimation of efficiency of the 18 context-category combinations that were the focus of our main RSA analysis. We did not optimize this design for the estimation of single-trials. Moreover, estimates of neural activity for single-trials in a rapid event-related design are heavily contaminated by adjacent trials (Mumford et al., 2012), which is especially problematic for this question where we would be interested in how the strength of a representation on a previous trial influenced reaction times on the next. In particular, we would expect that switching would be harder, and hence RTs longer, when the representation of the latent state was stronger on the previous trial (similar to an analysis of EEG data by Kikumoto and Mayr, 2020).

These points notwithstanding, we set out to examine if this test was feasible by carrying out a single-trial RSA analysis. We estimated a GLM for each trial in each participant, using the approach of Mumford et al., (2012), where a single regressor is created for the trial of interest in each GLM, and all other trials are modeled with regressors grouped by condition. We then extracted data from voxels in an ROI defined by the whole-brain RSA analysis (as above), and calculated cross-validated Mahalanobis distances between patterns on each trial and the average patterns for each of the 18 conditions.

Given that we did not optimize the design to obtain these effects, we were particularly concerned about backwards contamination of pattern activity on switch trials, such that the hemodynamic response on the trial where the latent state switched (trial n+0) might affect estimates of activity in the preceding trial (trial n-1). This would make it challenging to obtain an estimate of the strength of the latent state from pattern activity on trial n-1 that is not biased by trial n+0. To test this, we compared the cross-validated Mahalanobis distances of trial n-1 to the latent task state in trial n+0 for short and long inter-trial intervals (ITIs) – with the expectation that backward contamination would be worse for shorter ITIs. The analysis bore out this prediction, as this distance was significantly smaller on trials where the ITI was less than the median ITI (*t* = 2.30, *P* = 0.036, within-subject t-test). See Author response image 2 showing these distances for all 16 participants, each plotted as a single line. Having confirmed backward contamination from the hemodynamic response in the subsequent trial we would be very hesitant to draw conclusions from any analysis testing how the strength of pattern activity on trial n-1 influenced behavior on trial n. Similarly, forward contamination from the previous trial would certainly impact our ability to draw conclusions about how pattern activity on trial n+0 affects behavior on these switch trials.

**Author response image 2. respfig2:** 

iii) To further support that regions in the brain are representing latent states and tracking which state the participant is in, a univariate analysis that identifies regions that are active during latent state switches should be performed. For example, the authors could contrast activity immediately following a latent state switch vs. within a latent state (e.g. trial 1 vs. trial 4 w/in mini block). Is this response bigger when both the context and latent state switches?

We believe that this suggestion may have partially arisen from a misunderstanding of our experimental design. Conditions in the scanned portion of the generalization phase were presented in a pseudo-randomized order to optimize efficiency for a rapid event-related design, rather than presented in mini-blocks with several trials of the same consecutive type repeated in a row. Participants completed a mini-blocked segment of the generalization phase (similar to the training phases) prior to entering the scanner (this data is presented in Figure 2A-D and Figure 3). Please also see Figure 1—figure supplement 1 for further detail about our study design.

However, this did not prevent us from carrying out the suggested univariate contrast for switch costs related to the latent state. We ran a new GLM with condition regressors for repetitions and switches of category, context and latent state. Contrasting context switch trials where the latent state remained the same or switched, there were no voxels that survived correction for multiple comparisons or passed the peak threshold of *P* < 0.001, uncorrected.

While it may be somewhat puzzling that we do not observe a neural effect despite finding an effect of latent state switches in our behavioral analysis, it is possible that we are somewhat underpowered for this analysis. Again, we did not design trial ordering in the magnet to optimize efficiency to estimating switch vs. repeat differences. Thus, the latent state only remained the same in approximately 1/3 of trials, making switches far more common and expected —likely mitigating any neural switch cost effect.

In sum, the results of these posthoc analyses did not yield definitive results. However, in the pre-registration of our experimental design and analysis plan we focused on testing for a latent state representation based on an RSA of task conditions and did not power our experiment for individual differences analyses, or univariate contrasts between latent state repeat and switch conditions. We believe our main results still stand without a direct link between behavior and neural data in the scanner, and are an important contribution to understanding the network representing task structure for the purpose of generalization. Moreover, the computational modeling analyses described above that were prompted by the reviewer comments have deepened our understanding of how this latent state representation may be organized and relate to behavior. In the discussion of the revised manuscript, we note the need to draw a more direct connection between this neural representation and behavior, and the importance of addressing this question in future work in the Discussion section.

4) The latent task RSA analysis revealed a network of regions that is very similar to a network of regions representing "event schemas" across individuals (e.g. Baldassano et al., 2018; Chen et al., 2017). These results should be discussed in a broader context of how the brain represents structure more generally. The supplemental analysis is nice and shows the link to prior work but is not sufficient.

We appreciate and agree with the need for a broader comparison of our results to other data in the literature examining representations of task and event structure. To address the specific point regarding the correspondence of these networks, we sought to more formally characterize the overlap of our latent state RSA network with these prior results. A copy of the Pearson correlation coefficient map for between-participant pattern similarity during spoken recall from Chen et al., (2017) was graciously provided to us by the first author (Figure 3B in that paper). Comparison of the overlap of this statistical map with the statistical map from our RSA analysis revealed some overlap in the posterior medial cortex and inferior parietal lobule, but very limited overlap in lateral PFC or medial PFC.

Author response image 3 shows the latent state t-statistic map thresholded at *P* < 0.001, uncorrected in blue-green in order to be inclusive, and the Pearson correlation map thresholded at r ≥ 0.03 in red-yellow. Numbers above slices correspond to z-coordinate of slices in MNI space.

**Author response image 3. respfig3:** 

While interesting, we think the overlap of these networks is only moderate at most. The region of the posterior medial cortex with the maximum t-statistic in our sample was also fairly posterior and dorsal to that found by Chen et al. These regions of posterior medial cortex differ in their anatomical and functional connectivity with other parts of the brain: the more dorsal peak found in our analysis corresponding closer to a division of the precuneus with projections to rostrolateral PFC, and the peak from Chen et al. and corresponding closer to posterior cingulate cortex — with more connections in medial PFC (Margulies et al., 2009), which may relate to differences in the pattern of overlap in PFC subregions. The event schema results of Baldassano et al., 2018 appear to be similar to those of Chen et al., with results in posterior medial cortex being even more constrained to the posterior cingulate cortex (Figure 2 of that paper).In the revision, we discuss correspondence of our results to these data and others in the Discussion section. In particular, we believe that the regions implicated in this study correspond more closely to frontoparietal networks implicated in cognitive control and abstract stimulus invariant representations of tasks (also supported by our comparison to resting state networks from Thomas Yeo et al., 2011 in Figure 5—figure supplement 1). Networks identified as representing abstract structure related to schematic world knowledge, like those in the papers referenced by the reviewers, may be more involved in processing the congruity between events or items and a schema — while those implicated in the current experiment may be more involved in using this schema-level knowledge to control action selection and decision-making. We do not think these functions are necessarily completely segregated though. Overlap within these networks and anatomical connections bridging the two may prove to be important lines of communication between networks allowing schematic knowledge to be flexibly updated with new experience, and maintained for the purpose of control, during the performance of a task.

The authors do find evidence for "category" representations in a priori ROIs but do not really discuss what the interpretation of these results is. Why is the hippocampus and OFC representing category information during generalization? What does this mean for the field as a whole? For example, it is well established that hippocampus is essential for creating orthogonal representations that are used to disambiguate similar events. How do these results agree or disagree with this interpretation?

We are hesitant to draw strong conclusions from the finding of category information in hippocampal or OFC pattern activity. We do not believe that these results are surprising or particularly revealing about the function of these regions. OFC receives inputs from higher-order visual areas in inferior temporal cortex (Price, 2007), and has neurons tuned to stimuli like faces (Rolls et al., 2006). Past work examining pattern activity in OFC has found evidence for an animacy signal (Chikazoe et al., 2014), like our category regressor, and other work has found that stimulus categories like faces versus scenes, or food versus objects, can be decoded from OFC activity (McNamee et al., 2013; Pegors et al., 2015). We might speculate that the function of this category representation in OFC during our task relates to calculating the value of a decision based on the current context and/or latent state.

The finding of category information in hippocampus may be a bit more surprising for the reasons mentioned by the reviewers, but on consideration we do not think this contradicts conventional ideas about the function of the structure. Human hippocampal neurons have been observed to have selective responses to categories like faces and objects (Kreiman et al., 2000), and pattern activity in hippocampus carries information about visual categories that distinguishes stimuli like faces and scenes (Kuhl et al., 2012). Our results are consistent with these observations. We also do not think that our results are necessarily instructive as evidence for or against the claims about a hippocampal role in orthogonalizing events, per se, as the task did not place particular demands on separating events from each other. In general, the task demanded that participants pool information over multiple events and trial-unique stimuli to infer the values of each stimulus category in each latent state. Thus, while it would be helpful to have orthogonal representations of stimulus category, it would not be helpful to have distinct representations of each event in this task. It is possible, rather, that the hippocampus is using this category information in service of retrieving option values, but this is a speculative conclusion.

We have revised the portion of the Results where we report these effects in order to place these results in this context.